# Diverse paths to broadly neutralizing antibody escape among HIV-1 strains

Alex C. Stabell[1,2], Songhee Lee [2], Debby J. Park [2], Christy L. Lavine[3], Sebastian Mejia Espinosa[3], Viren A. Baharani[2,4], Rachel Patejak[2], Michel C. Nussenzweig [4,5], Marina Caskey [4], Michael S. Seaman[3], Paul D. Bieniasz [2,5] ✉ & Theodora Hatziioannou [2] ✉

Broadly neutralizing antibodies (bnAbs) are promising agents for HIV-1 treatment and prevention. However, the genetic barriers and mutational pathways to viral resistance, which can limit therapeutic antibody utility, remain poorly defined. Here we developed a medium-to-high throughput approach to determine the mutations that confer resistance to neutralization by the bnAbs 3BNC117 and 10-1074 currently in clinical development. We performed 7,776 parallel selection experiments to identify bnAb resistance mutations in 15 primary isolates that span global HIV-1 genetic diversity. There was substantial variability among HIV-1 isolates in the identity of mutations that conferred bnAb resistance. For 12 of 15 HIV-1 isolates, single amino acid changes could increase bnAb $IC_{80}$ to >10 µg ml$^{-1}$. Some 3BNC117 resistance mutations conferred resistance to additional bnAbs targeting the same or different epitopes, and unconventional escape mechanisms were occasionally encountered. These data provide a rationale for selecting bnAb combinations that are most likely to achieve treatment success.

Human immunodeficiency virus type 1 (HIV-1) continually evades autologous neutralizing antibodies[1–5], providing persistent and varying antigenic stimulation. In ~1–5% of people with HIV-1, this results in the development of rare broadly neutralizing antibodies (bnAbs), that can neutralize a large fraction of global HIV-1 isolates[6,7]. Therapeutic application of monoclonal bnAbs can suppress viraemia[8–10], in contrast to earlier antibody therapies with limited breadth[11–13]. The use of bnAbs in individuals whose viraemia was previously suppressed by conventional antiretroviral therapy (ART) has shown promise as a long-acting HIV-1 therapy[14–16]. In a few cases, bnAbs have apparently induced antiretroviral-free control of infection[17,18], potentially enabling remission.

Despite these promising results, resistance to bnAb neutralization remains a critical barrier to further clinical development[8–10,14–16,19–22].

Some clinical trials have used neutralization assays to evaluate viral reservoirs[18,23,24] and exclude participants with pre-existing bnAb resistance. Such approaches are reliable for certain bnAbs, for example, V3 glycan-dependent bnAbs[10,25–27], but unreliable for CD4 binding site bnAbs[15,27–29]. Comprehensive identification of bnAb resistance mutations has also been investigated using deep mutational scanning; however, the number of HIV-1 isolates that can be feasibly tested in this approach is limited[30–32].

Evaluation of bnAb escape across a panel of HIV-1 primary isolates that span global viral diversity has not been previously attempted and it is unknown whether there is a subtype/geographic variation in the propensity of HIV-1 to acquire bnAb resistance. We established a medium-to-high throughput assay, coupled with computational and experimental validation pipelines, to select and identify bnAb

[1]Weill Cornell, Department of Infectious Diseases, New York, NY, USA. [2]Laboratory of Retrovirology, The Rockefeller University, New York, NY, USA. [3]Center for Virology and Vaccine Research, Beth Israel Deaconess Medical Center, Harvard Medical School, Boston, MA, USA. [4]Laboratory of Molecular Immunology, The Rockefeller University, New York, NY, USA. [5]Howard Hughes Medical Institute, The Rockefeller University, New York, NY, USA. ✉e-mail: pbieniasz@rockefeller.edu; thatziio@rockefeller.edu

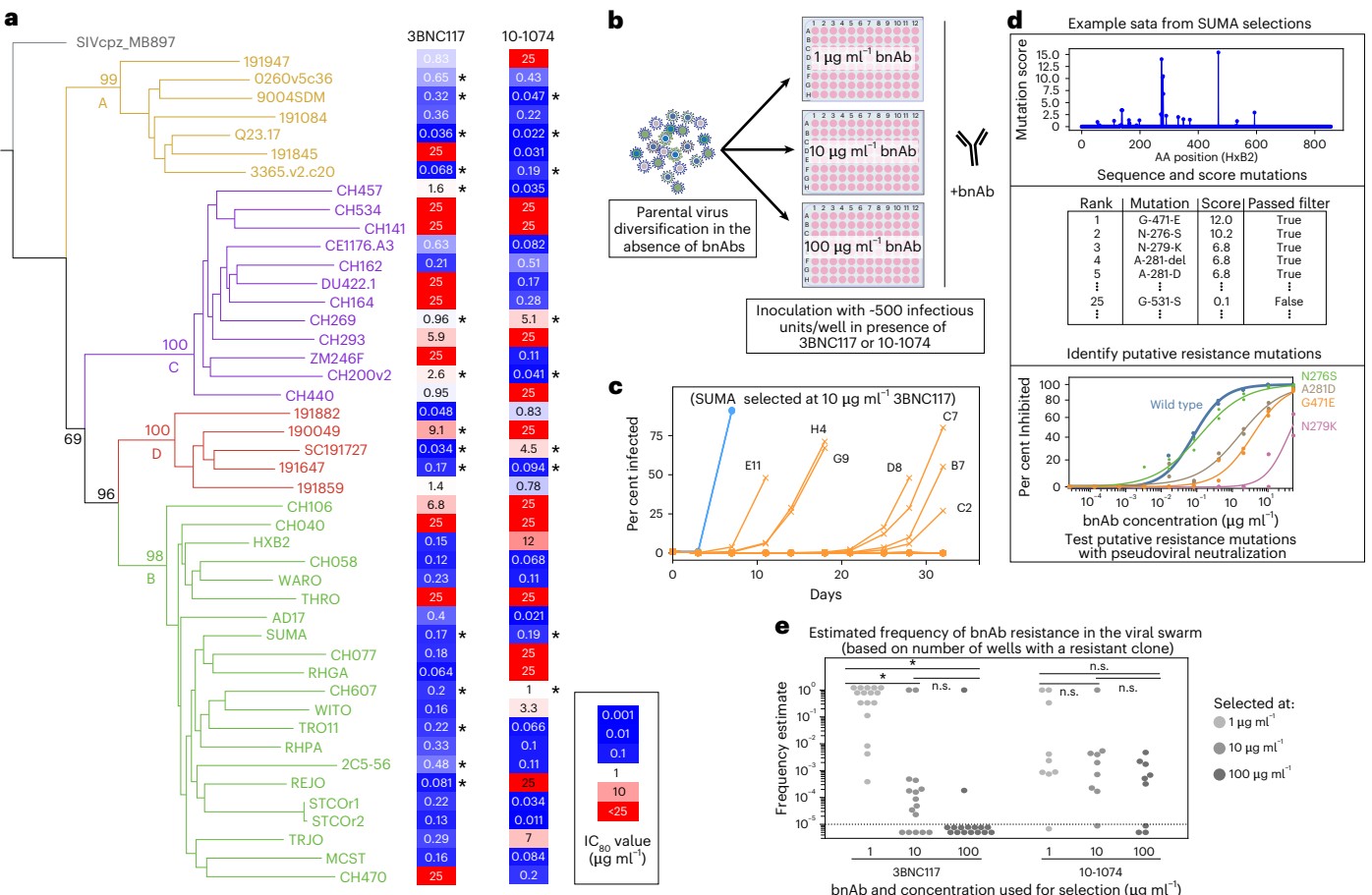

**Fig. 1 | Selection for 3BNC117 or 10-1074 resistance across a diverse panel of HIV-1 strains. a**, Phylogeny of *env* coding sequences from primary HIV-1 strains from subtypes A, B, C and D. The 3BNC117 and 10-1074 $IC_{80}$ value (in µg ml⁻¹) is indicated to the right of each strain. Viruses included in this study are indicated with asterisks. **b**, An overview of the experimental strategy used to identify putative 3BNC117 or 10-1074 resistance mutations. **c**, An example of data obtained with selection of the SUMA virus strain at 10 µg ml⁻¹ 3BNC117.

The blue line represents parental virus growth kinetics in the absence of 3BNC117 and the orange lines are resistant isolates emerging over time. **d**, An example of sequencing and phenotypic analysis of the SUMA strain as in **c**. AA, amino acid. **e**, The estimated frequency of bnAb-resistant variants for 3BNC117 and 10-1074 in the starting virus population before selection, derived from the number of starting viruses screened and the number of wells where virus replication was measured. *$P < 0.05$; n.s., not significant.

## Results

### Selection for 3BNC117 or 10-1074 resistance across a diverse panel of HIV-1 strains

To identify mutations that allow HIV-1 to escape bnAb neutralization, we performed selection experiments with HIV-1 strains from clades A, B, C and D. Fifteen viruses were chosen for 3BNC117 resistance and nine viruses for 10-1074 resistance (Fig. 1a). These viruses included 14 transmitted-founder isolates and one isolate (2C5-56) derived from a pretreatment plasma sample of a viraemic ART-naive individual enrolled in a clinical trial evaluating 3BNC117 administration[8]. Infectious molecular clones (IMCs), used to generate viral stocks, contained either full-length parental virus genome or were engineered by inserting the parental HIV-1 strain envelope gene (*env*) into an NL4.3-based IMC backbone (NHG)[34] (Supplementary Table 1). IMCs were transfected

into 293T cells, producing genetically homogeneous virus stocks that were then passaged in a T cell-based GFP indicator cell line (MT4-R5-GFP cells)[35] to introduce genetic diversity (Extended Data Figs. 1a and 2). Fresh MT4-R5-LTR-GFP indicator cells in each well of a 96-well plate received one bnAb and approximately 500 infectious units of the genetically diversified viral population, a number empirically determined to yield a single resistant virus per well after selection. Selection assays were performed in parallel at three antibody concentrations (1, 10 and 100 µg ml⁻¹) (Fig. 1b). Viral replication was monitored by measuring GFP fluorescence. Selection for bnAb resistance mutations was inferred when only a subset of wells in a 96-well plate showed evidence of viral replication (Fig. 1c and Extended Data Fig. 1a,b). Viruses from such wells were expanded in the presence of the same concentration of bnAb and resistance was confirmed through neutralization assays of the recovered isolates in comparison with the parental strains. Numerous individual isolates resistant to 3BNC117 or 10-1074 were derived from each of the parental viruses (Supplementary Table 1) and their complete *env* sequences were determined. During replication in cell culture, viral populations accumulate genetic diversity due to a combination of random genetic drift and cell culture adaptation[36,37]. For this reason, we developed a bespoke bioinformatic pipeline, which included a custom scoring heuristic for each mutation integrating both the frequency of mutations in the diversified parental virus population before selection and their enrichment in resistant isolates (Fig. 1d and

resistance mutations. Using 15 diverse HIV-1 strains and two clinically deployed bnAbs, the CD4-binding site antibody 3BNC117 and the V3 glycan-dependent antibody 10-1074, we conducted thousands of parallel selection assays and identified dozens of bnAb resistance mutations. We found substantial variability among HIV-1 isolates in their propensity to escape bnAbs, as well as in the identity of mutations, and mechanisms, that conferred bnAb resistance. With numerous clinical trials underway[33], this study expands the known landscape of bnAb escape mutations.

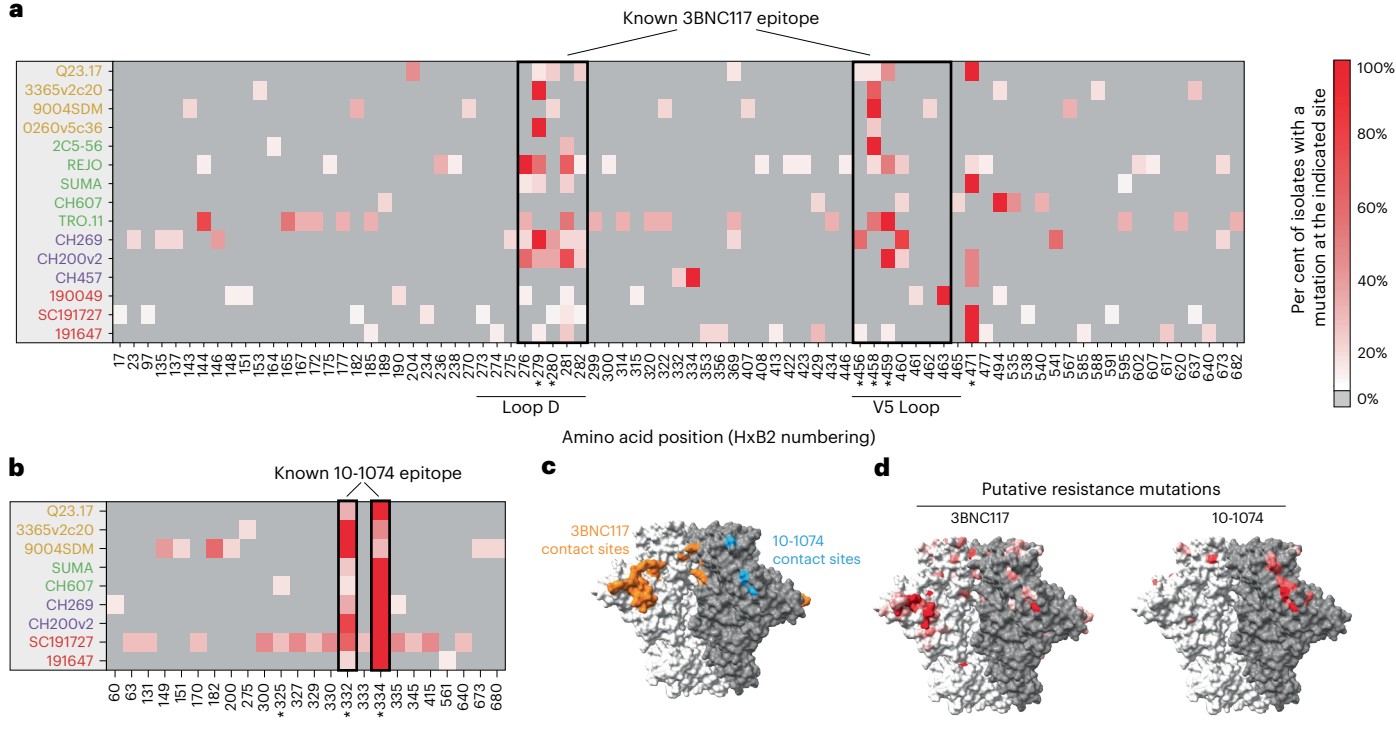

**Fig. 2 | Putative resistance mutations map within and outside known 3BNC117 and 10-1074 epitopes. a,b**, Putative bnAb resistance mutations as determined by short-read sequencing and bioinformatic analysis for isolates selected with 3BNC117 (**a**) or 10-1074 (**b**). The intensity of red at each Env position (HxB2 numbering) reflects the percentage of isolates had a mutation at each position for each parental strain. Asterisks indicate sites shown or inferred to be important for bnAb susceptibility by prior studies. **c**, Amino acids important for 3BNC117 and 10-1074 binding reported by prior studies are mapped on a crystal structure of the HIV-1 Env trimer (PDB 5v8m). **d**, Putative resistance mutations identified in this study mapped on the HIV-1 Env trimer coloured as in **a** and **b**.

Extended Data Figs. 3 and 4a). This site-agnostic approach prioritized mutations that were rare before selection but repeatedly observed in independent resistant viruses, consistent with positive selection under antibody pressure (Methods).

To estimate the total number of viruses in the diversified parental populations screened, limiting dilution assays were performed to accurately determine the number of infectious units per well for each strain tested (Extended Data Fig. 1a). This value, together with the number of wells where bnAb-resistant viruses emerged, provided an estimate of the frequency of bnAb-resistant viruses in the diversified starting population (Fig. 1e). For 3BNC117, higher bnAb concentrations reduced the measured frequency of resistant viruses, presumably reflecting heterogeneity in the magnitude of resistance exhibited by the resistant mutants. Most viruses were selected at 10 μg ml$^{-1}$ 3BNC117; however, two subtype C viruses (CH457 and CH200v2) required 100 μg ml$^{-1}$ for selection, suggesting a higher intrinsic resistance to 3BNC117. By contrast, for 10-1074, there was no significant difference in the frequency of resistant viruses for any bnAb concentration (Extended Data Table 1), consistent with the notion that 10-1074 resistance is a more binary 'all or nothing' phenomenon.

### Putative resistance mutations map within and outside known bnAb epitopes

Sequencing of 406 3BNC117-resistant isolates revealed 531 different mutations and 181 were algorithmically identified as possible escape mutations based on their low frequency in the unselected parental virus populations and enrichment in resistant isolates. Of these 181 mutations, 91 were located either within the known 3BNC117 binding epitope (as determined by the crystal structure)[38] or at a site previously identified by experimental or computational analysis as important for 3BNC117 susceptibility[18,26,29] (Fig. 2a). Changes within the 3BNC117

epitope were not sampled equally across clades. For example, clade D strains sampled loop D and V5-loop mutations less frequently than other subtypes, with the caveat that the number of strains tested for each subtype was low (Fig. 2a). For 10-1074 selection experiments, substitutions at positions 332, 334 or both were observed for isolates from all parental strains (Fig. 2b). Mutations at these sites—as well as at position 325—have previously been reported in individuals who developed resistance during 10-1074 therapy[10].

Some putative resistance mutations arose outside the 3BNC117 or 10-1074 binding epitopes (Fig. 2a,b). The 3BNC117-resistant TRO.11 isolates, in particular, acquired a large number of putative resistance mutations (24 putative sites identified across 19 resistant isolates) with no single obvious dominant mutation shared between isolates (Extended Data Fig. 3a). Similarly, 10-1074-resistant isolates derived from several strains, particularly SC191727, showed evidence of escape mutations outside of the known 325-332-334 (V3 glycan) epitope (Fig. 2b). Despite these exceptions, most putative resistance mutations tended to cluster at or near the known 3BNC117 or 10-1074 binding epitopes when mapped onto a HIV-1 Env trimer structure (Fig. 2c,d).

### Single amino acid substitutions confer resistance to 3BNC117 and 10-1074 across diverse HIV-1 strains

We tested each putative 3BNC117- or 10-1074-resistance mutation individually in a pseudotype neutralization assay using a set of parental virus Env expression plasmids (Fig. 3 and Supplementary Table 2). For 3BNC117, 59/181 candidate mutations conferred resistance, while for 10-1074 50/63 candidate mutations conferred resistance (resistance was defined as >10-fold increase in the 80% inhibitory concentration (IC$_{80}$) over wild type or an IC$_{80}$ >10 μg ml$^{-1}$; Fig. 3a,b). The lower positive predictive value of our scoring algorithm for 3BNC117 resistance mutations compared with 10-1074 resistance mutations may reflect a

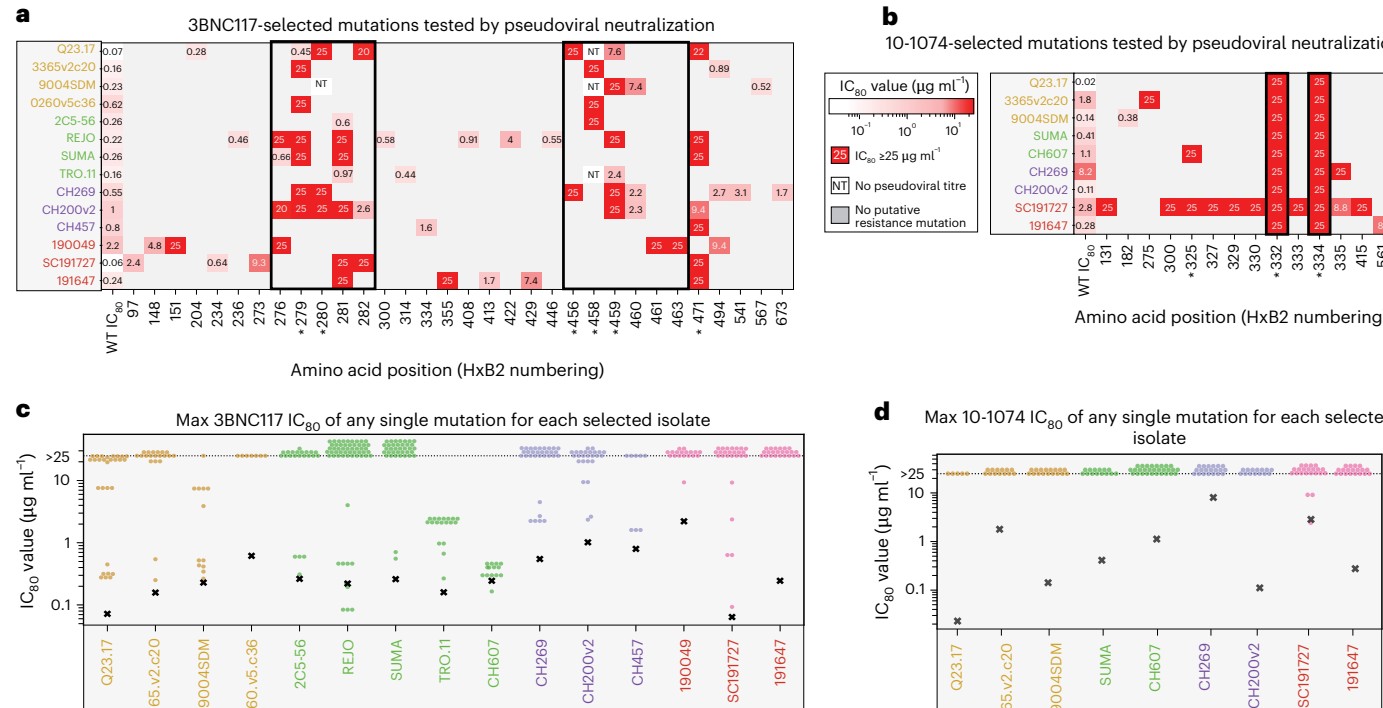

**Fig. 3 | Single amino acid substitutions confer resistance to 3BNC117 and 10-1074 across diverse HIV-1 strains. a,b,** $IC_{80}$ values for pseudoviruses containing Env with individual amino acid substitutions at the indicated amino acid position (HxB2 numbering) for 3BNC117 (**a**) or 10-1074 (**b**). WT, wild type. If multiple amino acid changes were identified at an individual site, the $IC_{80}$ value for the change that conferred the highest level of resistance is shown. Known bnAb epitopes are boxed. Asterisks indicate sites shown or inferred to be important for bnAb susceptibility by prior studies. **c,d,** Maximum $IC_{80}$ values obtained with individual amino acid changes for resistant isolates (circles) compared with the parental strain $IC_{80}$ (black x) for viruses selected for resistance to 3BNC117 (**c**) or 10-1074 (**d**). For each resistant isolate, only the single mutation that conferred the greatest increase in $IC_{80}$ in a pseudoviral neutralization assay is plotted.

lower abundance of 3BNC117 resistance mutations in the viral population before selection (Fig. 1e and Extended Data Fig. 4c,d), which in turn are more likely to be selected together with additional 'passenger' mutations. For 3BNC117, most (85%) of mutations conferring resistance resided in or around the known bnAb epitopes (Fig. 3a). At least one strain from each clade, and all strains in the case of clade D, acquired amino acid changes outside the known 3BNC117 epitope and CD4 binding site, although such changes generally conferred a lower level of resistance than those within the known epitope (Fig. 3a,c and Extended Data Fig. 5). For 10-1074, most (74%) confirmed resistance mutations were at the glycan sites 332 or 334. Notably, 10-1074 resistant isolates of the clade D strain SC191727 revealed multiple resistance mutations outside of the 10-1074 glycan epitope (Fig. 3b).

For 12/15 and 9/9 viruses tested, single amino acid substitutions were sufficient to confer resistance to 3BNC117 or 10-1074, respectively (Fig. 3c,d). Only 20 of the 406 3BNC117-resistant isolates contained more than one resistance mutation. There were three instances where single amino acid substitutions did not confer 3BNC117 resistance. The 3BNC117-selected CH607 isolates exhibited very low to no measurable cell-free infectious virus titre, despite evidence of viral spread in cells (Extended Data Figs. 1b and 5a). Moreover, introduction of the individual or combinations of selected mutations into the CH607 parental *env* did not change cell-free pseudotype susceptibility to 3BNC117 (Extended Data Fig. 6b,c). Rather, these mutations allowed for cell-to-cell spread[39–41] in the presence of 1 µg ml⁻¹ 3BNC117 (Fig. 4a). Spread to uninfected cells was eliminated by separating infected and uninfected cells by a transwell insert or by increasing 3BNC117 concentration to 10 µg ml⁻¹ (Fig. 4b and Extended Data Fig. 6d). Mutations identified to increase cell-to-cell spread resulted in an increase in the positive electrostatic charge of the CH607 Env surface (Extended Data Fig. 6e), possibly enhancing nonspecific cell membrane

attachment. In the case of the 9004SDM strain, introduction of the putative resistance mutations found in 9 of the 21 isolates resulted in very low-titre or non-infectious pseudoviruses (Extended Data Fig. 6f), suggesting that such mutations required additional compensatory changes for proper Env folding, expression or function. Nevertheless, we identified three alternative V5-loop deletions that conferred 3BNC117 resistance when introduced into the 9004SDM *env* (Fig. 4c). Finally, for the TRO.11 strain, no single mutation recovered from 19 resistant isolates conferred a high degree of 3BNC117 resistance in the pseudotype assay (Fig. 3a,c). Resistance appeared to require multiple amino acid substitutions with a progressive increase in 3BNC117 resistance as additional mutations were introduced (Fig. 4d). These exceptions highlight that while most bnAb-resistant phenotypes are driven by single amino acid substitutions, the genetic barrier to acquisition of 3BNC117 resistance appears higher for certain virus strains, requiring complex evolutionary mechanisms for escape.

### The 3BNC117 and 10-1074 resistance mutations are well sampled and context dependent

We next assessed whether 3BNC117 resistance conferred by single-residue mutations was context dependent. None of the substitutions were found in all viruses tested, but a subset were found in multiple viruses isolates (maximally 7/15 viruses). Most 3BNC117 resistance sites (65%) were uniquely identified in only a single viral isolate (Fig. 5a), suggesting that escape pathways are highly influenced by the Env context. A bootstrap resampling analysis of all 3BNC117-resistant isolates showed that the number of resistance sites increased with sample size but followed a log-power relationship, indicating diminishing returns with further selection experiments. This suggests our experiments captured the majority of all resistance sites in the panel of parental viruses tested (Fig. 5b). To directly test context dependence, we

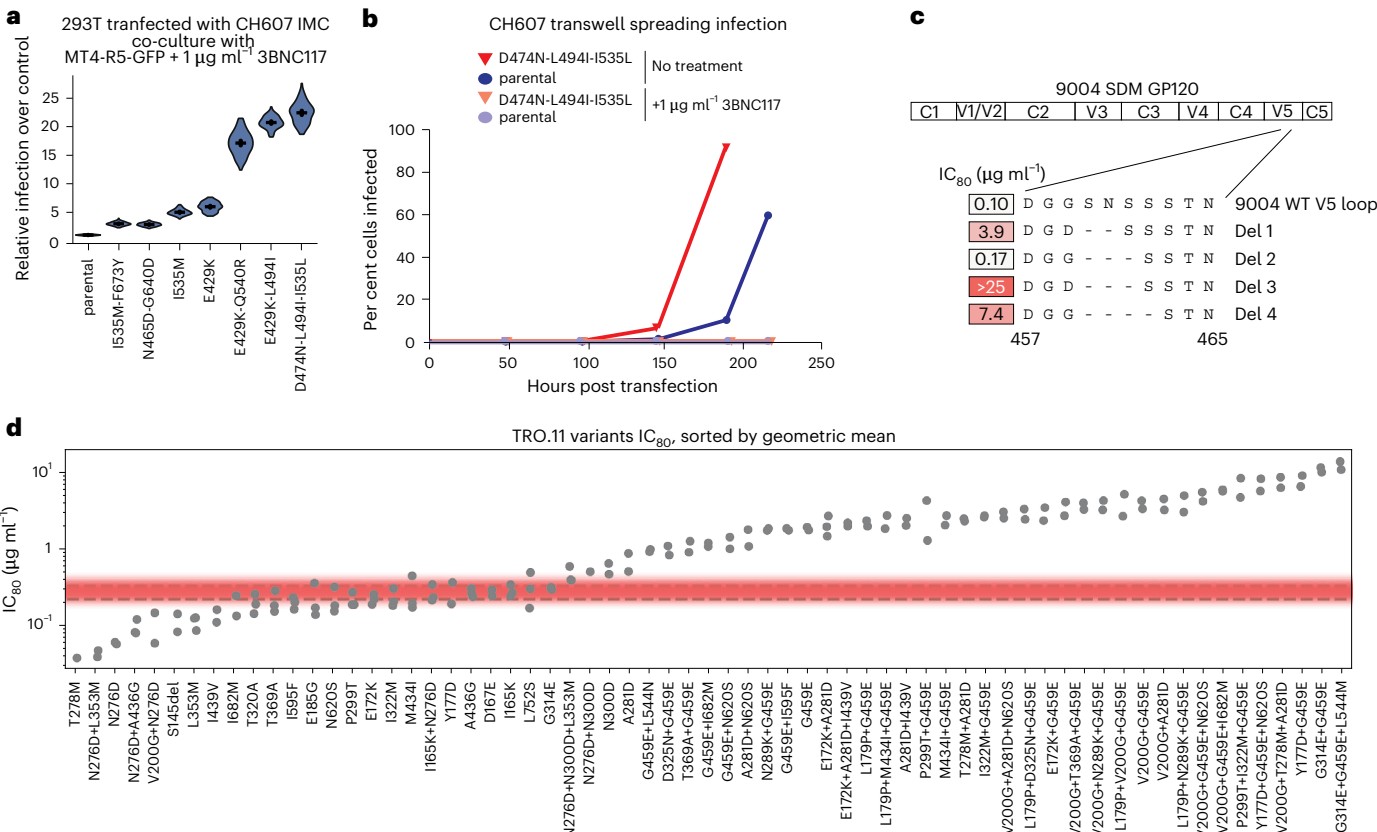

**Fig. 4 | Unusual mechanisms of 3BNC117 resistance in three HIV-1 strains involve enhanced cell-to-cell transmission or accumulation of multiple amino acid substitutions. a**, The 293T cells were transfected with an IMC of CH607 bearing the indicated mutation and co-cultured with MT4-R5-GFP cells in the presence of 3BNC117 at 1 μg ml⁻¹. Relative infection was calculated for each mutant and violin plots show the distribution of relative infection values generated from all replicate combinations (3⁴ = 81 per mutant; Methods). Horizontal bars indicate the bootstrap mean and error bars represent the 95% resampling confidence interval. **b**, The experimental set up as in **a**, except

293T and MT4-R5-GFP cells were placed in different compartments of a 0.2-μm transwell. **c**, The 3BNC117 IC₈₀ values against viruses pseudotyped with wild-type 9004SDM Env or those carrying the indicated V5-loop deletions selected in our experiments. **d**, The 3BNC117 IC₈₀ values against viruses pseudotyped with TRO.11 Env bearing the indicated mutations. The red gradient represents the range of values obtained for the wild-type TRO.11 Env over multiple experiment replicates. Each dot represents an independent experimental replicate for each indicated mutational combination.

introduced 45 resistance mutations that were uniquely identified in one viral context into three heterologous Env backgrounds (REJO, 2C5-56 and CH269; Supplementary Table 3). Only two of these mutations (4%) conferred resistance in the heterologous background, confirming that most 3BNC117 resistance mutations are strain specific (Fig. 5c). Notably, 7 of the 45 mutants tested failed to produce infectious pseudotypes in the heterologous Env context, suggesting that some mutations carry a high context-dependent fitness cost and may be non-viable owing to functional constraints (Supplementary Table 3).

For 10-1074, the SC191727 strain acquired multiple unique mutations outside the canonical V3 glycan epitope (Fig. 3b), accounting for nearly all non-glycan resistance sites identified in our dataset (Extended Data Fig. 7a). We introduced each of these changes into three heterologous Env backgrounds, Q23.17, SUMA and CH269. Resistance conferred by these mutations was again largely context specific. Only 2 of 17 SC191727 mutations conferred resistance in Q23.17 or SUMA (Extended Data Fig. 7b). Moreover, both involved position 325, a site previously associated with 10-1074 resistance in clinical studies[10]. CH269, which had a high baseline IC₈₀ (ranging from 5 to 8 μg ml⁻¹), showed modest increases in resistance when SC191727 mutations were introduced (Extended Data Fig. 7b). This suggests that CH269 was already near a resistance threshold. However, during our selection experiments, CH269 favoured glycan modifications, which conferred increased resistance. Overall, these data suggest that our assays

identified most resistance mutations available to the viruses studied for both bnAbs, and mutations affecting the protein backbone are highly context dependent in their effects on bnAb sensitivity and viral fitness.

## Selection for 3BNC117 resistance can cause resistance to additional bnAbs

We tested each 3BNC117-resistant mutant pseudotype against a panel of antibodies that also targeted the CD4 binding site (N6, VRC07.523, VRC01, VRC01.23, 04-A06 and b12) and antibodies targeting the V2 glycan (PGDM1400 and PG9) and V3 glycan (10-1074). Most 3BNC117 resistance mutations did not confer cross-resistance. Instead, many of these mutations increased neutralization sensitivity to other bnAbs, as reflected by lower IC₈₀ values compared with the parental virus (Fig. 6a and Supplementary Table 4). Despite this general pattern, for each CD4 binding site bnAb at least one 3BNC117 resistance mutation could cause cross-resistance (Fig. 6b, Extended Data Fig. 8a and Supplementary Table 4). In the case of VRC01 and VCR07.523, several 3BNC117 resistance mutations conferred cross-resistance. Nearly all 10-1074 resistance mutations conferred cross-resistance to PGT121 (Extended Data Fig. 9a and Supplementary Table 4), a clonally related antibody that targets the same glycan epitope[42]. Otherwise, little or no cross-resistance to additional bnAbs targeting the CD4 binding site (3BNC117 and VRC07.523) or the V2 glycan (PGDM1400) was observed (Extended Data Fig. 9b).

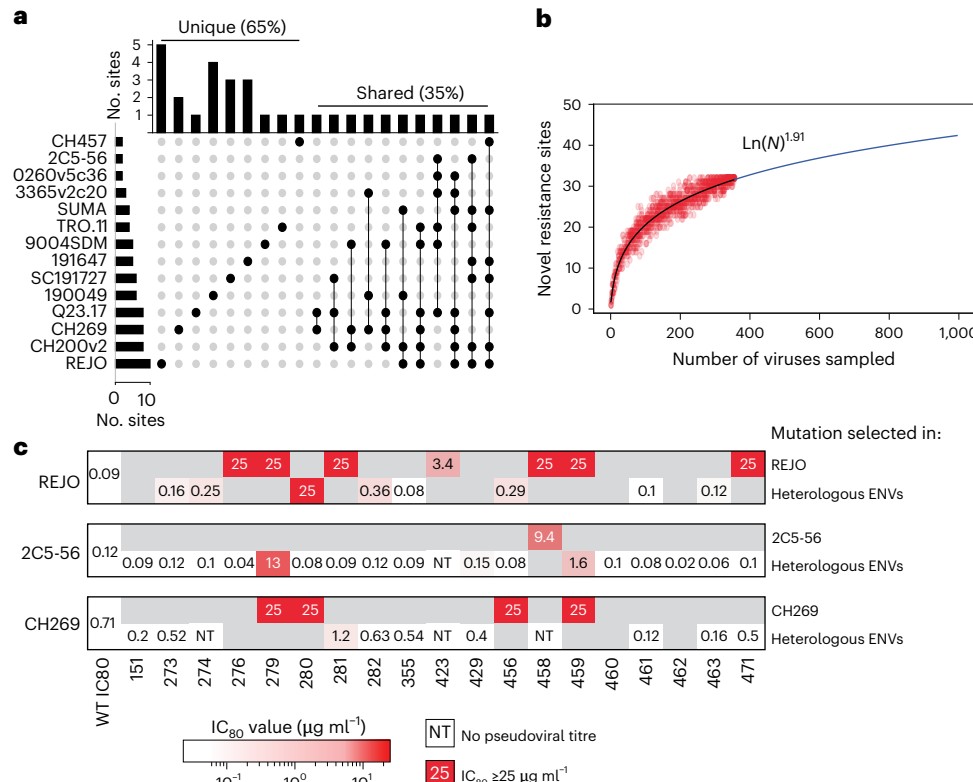

**Fig. 5 | 3BNC117 resistance mutations are well sampled and context dependent. a**, An UpSetPlot of 3BNC117 resistance sites tested. Each row represents a viral Env and each column represents a unique combination of resistance sites (dot matrix). The top bar graph shows how many sites fall into each combination, while the side bar graph shows how many resistance sites were identified in each Env. **b**, Bootstrap resampling of the sequencing data showing the number of resistance sites detected for the number of antibody resistance viral isolates sequenced. **c**, The 3BNC117 IC$_{80}$ values (µg ml$^{-1}$) for pseudoviruses with Env bearing mutations either selected in the context of the strain indicated (top) or swapped from a heterologous viral Env (bottom).

## Discussion

Our findings demonstrate that the genetic barrier to resistance to two clinically deployed bnAbs is variable and strain dependent. The genetic barrier for evasion is low for the majority of virus strains, with a single mutation conferring escape from each bnAb. Individual escape mutations can arise within and outside the known antibody epitope, although for a few strains more complex pathways to resistance were identified. While there was overlap in the pathways to bnAb escape among the different HIV-1 strains, numerous strain-specific strategies were also evident among this diverse group of viruses.

Consideration of HIV-1 population sizes, polymerase error rates and the nature of bnAb resistance mutations indicates that variants that are resistant to an individual bnAb will be present in most viral populations, but these variants become evident and quantifiable only after selection. Estimates based on our selection assay reveal that, in a diversified, unselected population, resistant variants across multiple viral strains are detected at a frequency of roughly 10$^{-2}$ to 10$^{-4}$ for 10-1074 and 10$^{-3}$ to 10$^{-5}$ for 3BNC117, (resistance defined as IC$_{80}$ >10 µg ml$^{-1}$). Assuming that these frequencies are similar in unselected populations in vivo, it is likely that most individuals, even those treated with ART, would harbour bnAb-resistant variants given that a typical latent reservoir size is about 1–100 per 10$^{6}$ CD4$^{+}$ T cells[46] and there are around 10$^{11}$ CD4$^{+}$ T cells in a person[47]. Assessing bnAb resistance during clinical trial enrolment[33] is thus inherently limited by the small fraction of the viral reservoir that can be sampled. A more illuminating, albeit labour intensive approach may be to infer the likelihood of resistance emergence by testing a small number of representative isolates from each individual and quantifying the frequency with which bnAb-resistant variants arise.

Mutations conferring cross-resistance to 3BNC117 and other CD4 binding site bnAbs were not randomly distributed but clustered within specific structural elements of the CD4 binding site. Such mutations were enriched in loop D, β20 element or the V5-loop (Fig. 6b). Substitutions in loop D affected susceptibility to N6, VRC07.523 or VRC01.23 whereas substitutions in both loops D and V5 reduced susceptibility to VRC01. Notably, 04-A06 susceptibility was not impacted by mutations in the canonical CD4 binding epitope, and only mutations in the β20 element conferred resistance to this bnAb. Notably, β20 element mutations also conferred resistance to V2 glycan bnAbs (PG9 and PGDM1400), suggesting that such mutations promoted a more open trimer conformation (Fig. 6c,d). This inference is supported by some studies showing that an open Env trimer conformation can confer resistance to a number of neutralizing antibodies[43–45], although this mechanism of neutralization resistance is uncommon among primary isolates (Fig. 6e).

Before this study, knowledge of bnAb escape pathways was limited to a handful of HIV-1 strains that did not represent the global virus diversity[26,28–32,52,53] or narrowly focused on natural history cohorts[28,48–51]. Efforts to map bnAb escape mutations using laboratory-generated mutagenesis[29,30,52,53] are limited by the small number of strains proven to be tractable to library generation. Such approaches only probe effects of single amino acid substitutions and miss combinatorial effects of multiple substitutions, deletions and insertions. Our methods overcome these limitations by relying on experimental evolution via virus-driven mutagenesis to identify resistance pathways, more accurately mimicking the mutational space sampled during virus replication in vivo. Coupled with our site scoring algorithm that improves the prediction of bonafide resistance mutations (Extended Data Fig. 4c,d), our approach allows the analysis of a significantly larger number of strains and the identification of unconventional escape pathways.

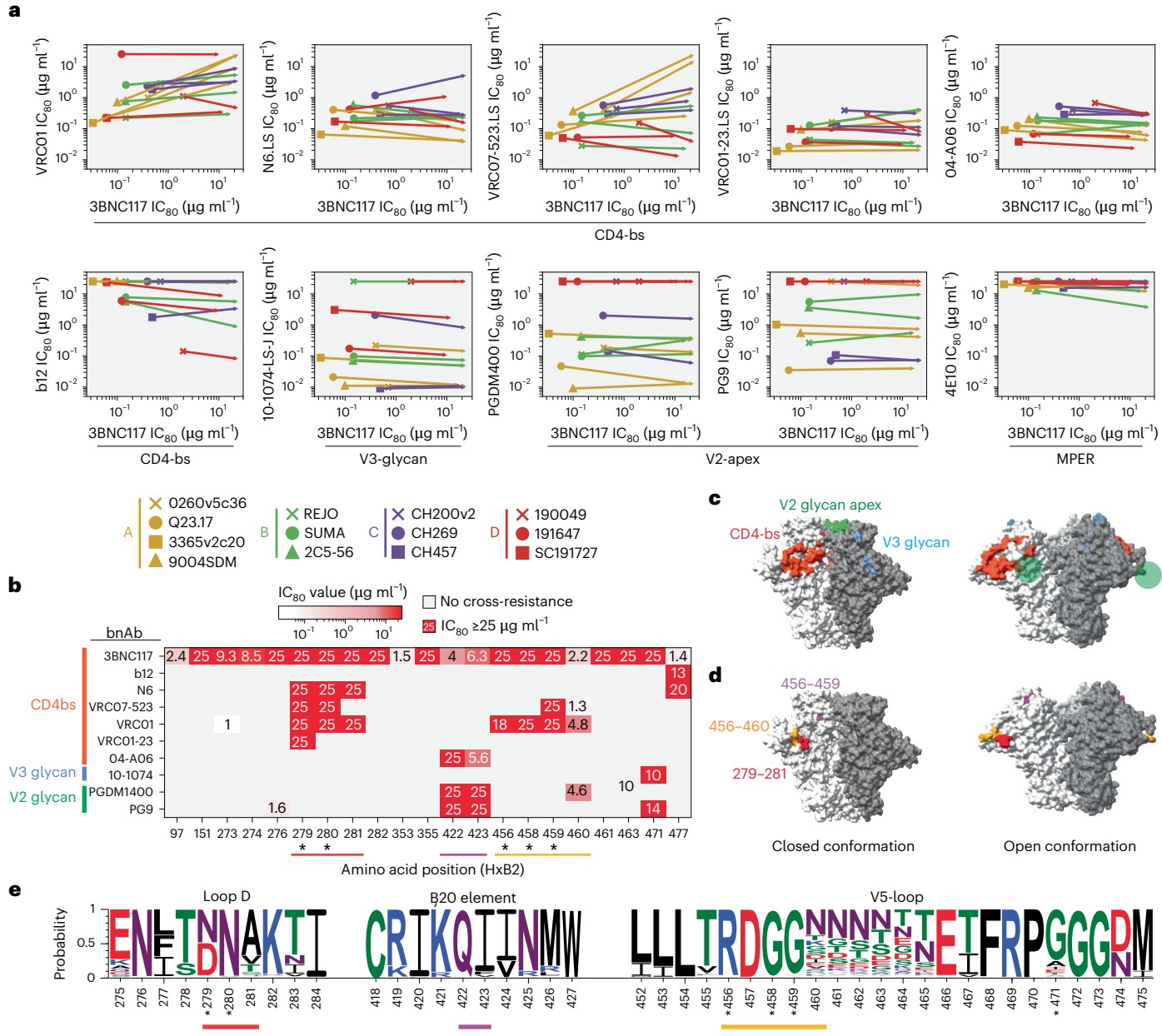

**Fig. 6 | Selection for 3BNC117 resistance can cause resistance to additional bnAbs. a**, Markers indicate the $IC_{80}$ values for 3BNC117 ($x$ axis) and a second antibody ($y$ axis) for each parental Env. The arrows represent the average change in $IC_{80}$ values due to 3BNC117 resistance mutations (geometric mean difference from the parental virus). **b**, Amino acid positions (HxB2 numbering) of 3BNC117 resistance mutations that induced cross-resistance to the bnAbs indicated on the $y$ axis. The asterisks indicate previously described sites important for 3BNC117 sensitivity. **c**, bnAb epitopes on the CD4 binding site (red), V3 glycan (blue) and V2 glycan (green) are shown on the closed (PDB 5v8m) or open (PDB 6u0l) Env trimer structure. The V2 glycan is not resolved in the open structure and its inferred location is indicated by green circles. **d**, Positions where changes confer cross-resistance to 3BNC117 and an additional bnAb are marked on closed and open conformations of the Env trimer structure. **e**, Logos plot of epitopes that conferred resistance to 3BNC117 and an additional bnAb. The probability of finding the indicated amino acid at each position (HxB2 numbering) in the Los Alamos HIV-1 sequence database is plotted ($y$ axis).

For example, while the majority of 10-1074 escape mutations identified herein are concordant with prior studies[10,30,32], selection of 10-1074-resistant clade D primary isolates revealed previously unknown resistance substitutions (Fig. 3b). Notably, targeted mutagenesis and deep mutational scanning have failed to identify mutations that confer a high degree of resistance to CD4 binding site bnAbs such as 3BNC117 and 04-A06 (refs. 29,30,32,53). By contrast, our approach identified multiple individual amino acid changes that confer near-complete resistance to CD4-binding site targeting bnAbs in most viral strains (Figs. 3 and 6), as well as unusual mechanisms of escape (Fig. 4).

As with small-molecule antiretroviral drugs, elevating the genetic barrier to acquisition of resistance is an important goal and will necessitate the use of bnAb combinations. Some of the resistance mutations in this study were highly specific to the selecting antibodies, while others were able to confer resistance to at least one other bnAb targeting an overlapping epitope. Of interest were mutations in the $\beta20$ element that resulted in resistance to antibodies targeting both the CD4 binding site and other epitopes (Fig. 6). The $\beta20$ mutations appear to shift the Env to an 'open' state, which has been associated with increased neutralization by autologous antibodies and antibody-dependent

cellular cytotoxicity[45,54,55]. Such pathways of escape are thus unlikely to be favoured in vivo. Consequently, deployment of bnAb combinations for which the only escape pathway available leads to an 'open' Env conformation would increase the barrier to escape. Our study suggests practical approaches for prioritizing such bnAb combinations, for example, in the panel of viruses analysed, simultaneous escape from two of the CD4 binding site targeting bnAbs, 3BNC117 and 04-A06, was only found for changes that affect overall sensitivity to neutralization. Selection experiments with additional bnAb candidates in clinical development, such as 04-A06, would be necessary to exhaustively explore cross-resistance patterns. The ability of bnAb combinations to suppress acquisition of resistance could also be tested with our selection methods. In addition, determining the sensitivity of such putative escape viruses to neutralization by sera samples from infected individuals, before treatment with antibodies, could validate our conclusions and possibly provide a useful screening tool for bnAb trial participants.

Overall, this study establishes approaches to identify previously unappreciated pathways of bnAb escape and demonstrates that resistance to individual bnAbs can be easily acquired. This result indicates that bnAb breadth is not due to an intrinsically high barrier to resistance but rather to their rarity, that is, they are insufficiently common to have exerted selective pressure on the vast majority of circulating HIV-1 strains. A corollary of this conclusion is that a wide injudicious bnAb deployment will eventually compromise their efficacy. Nevertheless, approaches such as those documented herein can identify combinations of bnAbs with a higher barrier to escape that could guide strategies to minimize the development of resistance.

## Methods

### Phylogenetic analysis
The sequences for each HIV-1 *env* were courtesy of Michael Seaman. Full-length *env* HIV-1 sequences were aligned using the muscle ppp algorithm. A phylogenetic tree with bootstrap values was estimated with RaxML using a general-time-reversible model with invariable sites using the following command 'GTRGAMMAI -f a -x 1 -N 100 -p 1'. The tree was visualized in TreeViewer. The $IC_{80}$ values listed were obtained in the current study, provided by Michael Seaman or obtained via the catnap database.

### Generation of IMCs
Full-length IMCs for 9004SDM, REJO, SUMA, CH607, CH457, CH200v2, CH269, 190049, 191647 and SC191727 were provided by Michael Seaman. Env expression vectors for Q23.17, 3365v2c20, 0260v5c36 and TRO.11 were provided by Michael Seaman. The 2C5-56 *env* was synthesized via addgene in geneblocks (2×1,500 bp each) using the reference sequence KX028084 (NCBI nucleotide database). Isolates that were not available as an IMC were cloned into the full-length HIV-1 backbone NHG (NCBI accession JQ585717.1) modified to include a 9-bp segment containing a NOT1 site (GCGGCCGCA) after the start codon of the eGFP gene that replaces NEF (referred to NHG-NOT1). NHG-NOT1 was cut with EcoR1 and NOT1 and gel purified using Macherey–Nagel NucleoSpin Gel and PCR clean-up kit (740609). A 500-bp fragment used for assembly was PCR amplified from NHG-NOT1 using the primers ACS136F (5′-GCAGGAGTGGAAGCCATAATAAG-3′) and ACS137R (5′-TGCCACTGTCTTCTGCTCTTTC-3′). This 500-bp fragment, the *env* of interest, and the NHG-NOT1 backbone were assembled using Takara in-fusion cloning master mix (638945) using the manufacturer's specifications except for at half the recommended volume. For CH607 cell-to-cell transmission experiments, *env* genes were amplified from pro-viral DNA and cloned into the NHG vector as described. All primers were purchased through Integrated DNA Technologies (IDT).

### Cell lines
HEK293T cells (ATCC CRL-3216) were maintained in Dulbecco modified Eagle medium (DMEM) supplemented with 10% fetal calf serum.

MT4-R5-GFP cells were derived from MT4-GFP cells[56], which were derived from MT4 cells (AIDS reagent program ARP-120, now through BEI). MT4-R5-GFP cells were derived from MT4-GFP cells by transducing the original MT4-GFP cells with a virus stock made with CCR5 R126N in LHCX (cloned in the parental LHCX via HindIII and HpaI) packaged with MLVGagPol and VSV-G. Cells were selected with hygromycin at 25 µg ml⁻¹ and single-cell cloned by limiting dilution. Cells were maintained in RPMI 25 µg ml⁻¹ hygromycin and 1.25 µg ml⁻¹ puromycin. MT4-R5-GFP-CD4high lines were generated through retroviral transduction of MT4-R5-GFP cells with an MLV-based vector system expressing CD4-IRES-DHFR. After transduction, 25 nM methotrexate was added to the media and single-cell clones were isolated and characterized. One clone (3B5) was used for further studies.

### Selection for bnAb-resistant virus
Infectious virus was generated by transfecting a 10-cm dish of 293Ts at 80% confluency. Then 10 µg of a plasmid containing an IMC (generation described above) was mixed with 40 µl PEI in 1 ml serum-free DMEM. This mixture was incubated at room temperature for 15 min and then added to the medium. Media were replaced 8–12 h after transfection and virus was collected 48 h later. MT4-R5-GFP or MT4-R5-GFP-CD4high cells (herein referred to MT4 cells) were infected at an multiplicity of infection of ~0.01 and virus was allowed to spread until >80% of cells were infected or >30% of cells in the media had died. The supernatant from this first passage was used to seed a second passage infected at a multiplicity of infection <0.1. When >80% of cells were infected or >30% of cells in the media had died, the cells were diluted to $5 \times 10^5$ infected cells per ml and diluted into fresh MT4 cells in the presence of either 3BNC117 or 10-1074 at 1, 10 or 100 µg ml⁻¹. The goal was to achieve an infected:uninfected MT4 cell ratio of 1:100 (that is, 500 infected cells per 200 µl + 49,500 uninfected cells per 200 µl). This cell suspension was plated in each well of a 96-well plate. Similarly, 5 ml of 500 infected cells per 200µl + 49,500 uninfected cells per 200 µL cell suspension was fivefold serially diluted and plated without the addition of an antibody to estimate the actual number of infected cells per well and is referred to as the 'control plate'. Viral infection was monitored by FACS and/or by visual inspection for a GFP signal. If there was spreading infection in all wells of a plate that contained 3BNC117 or 10-1074, the supernatant was collected and transferred to fresh MT4 cells in the presence of the bnAb at 10× higher concentration (that is, if spreading occurred in all wells at a bnAb concentration of 1 µg ml⁻¹, the supernatant was passaged to fresh cells at bnAb concentration 10 µg ml⁻¹). Selection for bnAb resistance was inferred when there were some wells with no viral replication and others with viral replication. Virus was collected from these selected wells when the per cent of cells infected was ≥40% and expanded in the presence of the bnAb used for selection. This culture was monitored visually until >80% of cells were infected or >30% of cells in the media had died and both the supernatant and cells were collected. The viral supernatant and cells were collected and frozen at −80 °C for further analysis. The same process was applied to control plates and a subset of viruses from individual wells from such plates were isolated.

### Estimation of bnAb resistance in the unselected viral population
The control plates described above were used to more accurately estimate the number of infected cells in each well of the 96-well plate. The number of wells that had >10% of cells infected before the end of each selection experiment (typically ~30 days) were called as 'positive'. The frequency at which resistant viruses existed in the starting viral population before selection was estimated by fitting *p* to the log likelihood function

$$L(p,k) = k\mathrm{Log}(p) + (96-k)\mathrm{Log}(1-p).$$

Where *p* is the probability that a virus has a bnAb resistance mutation and *k* is the number of wells that virus was replicating in a 96-well plate during selection.

## Illumina library preparation and sequencing

Pro-viral DNA was extracted from cells using QIAamp DNA mini kit (51304). Pro-viral DNA was amplified using Primestar GXL with ACS92 (5′-GGCTTAGGCATCTCCTATGGCAGGAAGAA-3′) and ACS93 (5′-GTTCTGCCAATCTGGGAAGAATCCTTGTGTG-3′). The PCR product was gel purified using the Macherey–Nagel NucleoSpin Gel and PCR Clean-up kit (740609). DNA purity was assessed by a Nano-drop. If the A260/230 ratio was <1, the DNA was cleaned up with isopropanol precipitation. All DNA was diluted to 0.2 ng μl$^{-1}$ in a 96-well plate and used as a template for Illumina-based library preparation using the Nextera XT DNA library preparation kit (FC-131-1096) and Nextera XT Index v2 Set A barcodes (FC-131-2001). Library preparation was performed to the manufacturer's specification with the following changes: volume of each sample was reduced by half and AMPure XP (Beckman A63881) was used instead of illumina purification beads. Libraries were sequenced on a MiSeq with a MiSeq Nano v2 flowcell (MS-103-1001).

## Sequencing data analysis and identifying likely bnAb resistance mutations

bwa-mem[57] was used to aligned fastq files to the full-length HIV-1 genomes used in this study. Variants were called using lofreq3[58] (used primarily for defining indels) and VirVarSeq[59], which call variants at the codon level. The following heuristic was developed to help identify the most likely bnAb-resistant viruses. For each amino acid substitution, the following score was calculated:

$$\text{Score} = -n \log f_c.$$

Where $f_c$ is a function of the frequency of the amino acid substitution in the pre-selection population and $n$ is the number of bnAb-resistant isolates in a single selection experiment that had this amino acid substitution. All extracellular regions of each Env were included in the analysis and the average read depth was 930 reads per nucleotide. To identify variants, software (lofreq and VirVarSeq) that take into account both depth of coverage and quality of reads were applied to the data. Amino acids were scored in bnAb selected variants only if their frequency was at least 25%. Variant frequencies in the control (non-bnAb selected) control populations as determined by VirVarSeq were used to score each amino acid substitution present in the selected isolates. Indels were scored using indel frequency calculations from lofreq. We note that the mutations identified in viruses isolated from the control plates did not differ from those obtained from the original diversified population before selection and sequencing data from both sources were combined to generate site scores. The top scoring amino acid substitutions were identified for each bnAb-resistant isolate until the sum of the frequencies exceeded 90%. In most cases, only a single amino acid was identified; however, there were instances where there was evidence of two or more distinct viral populations (for example, two putative resistance mutations at 50% frequency each).

## Cloning of *Env* expression vectors

*rev*/*env* were amplified from the IMCs for 9004SDM, REJO, SUMA, CH607, CH457, 190049, 191647 and SC191727 using the primers env1A (5′-CACCGGCTTAGGCATCTCCTATGGCAGGAAGAA-3′) and env1M (5′-TAGCCCTTCCAGTCCCCCCTTTTCTTTTA-3′). All but 9004SDM were cloned into pcDNA3.1dTOPO expression kit (Thermo K490001). 9004SDM was cloned into pBRR322 along with a flanking CMV promoter/enhancer and poly-A signal due to toxicity in bacteria when cloned into pcDNA3.1. The CH200v2 did not generate pseudovirus when cloned using this strategy so all neutralization assays were done on the full-length IMC with single amino acid substitutions introduced into the *env*.

## Site-directed mutagenesis

The target mutation was introduced into each Env expression vector using primers with 20–30 bp overlapping on each side of the target sequence. The entire plasmid backbone was amplified using 25-μl reaction of KoD-Xtreme Hot Start (Sigma 41105500) for 18 cycles. The parental plasmid was digested using 1 μl DPNl added directly to the PCR reaction and incubated at 37 °C for 2 h. The reaction was isopropanol-precipitated and 1 μl was used for transformation into STBL3 cells (Thermo C737303). Individual clones were sequence-confirmed using whole-plasmid sequencing (https://www.plasmidsaurus.com).

## Pseudoviral production

First 750 ng Env expression vectors and 1.5 μg SG3DEnv backbone plasmid were PEI transfected (using a 4:1 PEI:μg DNA ratio) into 293T (ATCC) in 12-well dishes. Media were replaced 8–12 h after transfection and the virus was collected 48 h after transfection, clarified and diluted 1:2 for use in neutralization assays in MT4 cells. For experiments testing cross-resistance, HIV-1 Env pseudovirus stocks were generated by transfection of 293T/17 cells (ATCC) in T75 flasks with 4 μg HIV-1 Env expression plasmid and 8 μg SG3DEnv backbone plasmid (NIH HIV Reagent Program) using Fugene transfection reagent as recommended by the manufacturer (Promega). Virus-containing supernatants were collected 24 h following transfection, filtered (0.45 mM) and stored at −80 °C in 1.0-ml aliquots. Virus stocks were titrated in TZM.bl cells (NIH HIV Reagent Program) to determine an assay use dilution that resulted in approximately 150,000 relative luminescence units.

## Neutralization assay

First 25 μl of the viral supernatant was mixed with 75 μl of RPMI with the desired antibody concentration. This was incubated for 20 min at room temperature followed by the addition of 100 μl of CD4-high MT4-R5-GFP cells at the concentration of $5 \times 10^5$ per ml. The highest antibody concentration in this assay was 50 μg ml$^{-1}$. Ten serial five-fold dilutions were used to generate a neutralization curve. At 48 h after infection, GFP-positive cells were counted via flow cytometry and a sigmoid neutralization curve was fit using the scipy library. For experiments testing cross-resistance, the TZM-bl neutralization assay was performed as previously described[60]. Briefly, mAbs were serially titrated in duplicate wells using a primary concentration of 25 μg ml$^{-1}$ and a fivefold dilution series. Following the addition of HIV-1 Env pseudovirus, plates were incubated for 1 h at 37 °C, followed by the addition of TZM-bl cells ($1 \times 10^4$ per well) in the presence of 11 μg ml$^{-1}$ DEAE-Dextran (Sigma). Wells containing cells + pseudovirus (without sample) or cells alone acted as positive and negative infection controls, respectively. Following a 48 h incubation, plates were collected using Promega Bright-Glo luciferase reagent and luminescence detected with a Promega GloMax luminometer. Titres are reported as the concentration of mAb that inhibited 50% or 80% virus infection (IC$_{50}$ and IC$_{80}$ titres, respectively).

## Bootstrap resampling to estimate potential increase in resistant sites identified with increased sampling

All sequencing data from each viral isolate characterized from the selection experiments was bootstrap resampled 1,000 times for 1, 2, 3,… and so on isolates using the numpy package in Python. For each number of resampled isolates, the number of confirmed bnAb resistance mutations (based on confirmed pseudoviral neutralization) was calculated and plotted. These data were modelled with linear, log, power-log and general-log functions. General-log function (that is log($N$)$^{a+b}$) had the best test of fit by Akaike Information Criteria and is reported.

## Cell-cell transmission experiments

The 293T cells were plated in a six-well dish and transfected at 80% confluency with 500 ng pcDNA expressing mCherry alone or 1 μg IMCs of CH607 containing the mutations identified as well as a 500-ng pcDNA vector expressing mCherry. All IMCs, including the control vector, contained a 9-bp deletion in the V1 loop that was present before

selection with 3BNC117 (ATGAGAGTA starting at codon 190). The control CH607 plasmid contained a short deletion in the V1 loop which was fixed in the parental virus before selection. At 8–12 h post transfection the cells were washed with PBS and trypsinized. Then 1,000 CH607 + mCherry transfected cells were plated with 50,000 MT4-R5-GFP cells in a 96-well plate. GFP positivity was quantified by flow cytometry. The relative infection $R_i$ was calculated by taking the following ratio:

$$R_i = \frac{m_{Ab}/m_{nt}}{C_{Ab}/C_{nt}},$$

where $m_{Ab}$ is the percentage GFP signal of the mutant in the presence of 3BNC117, $m_{nt}$ is the percentage GFP signal of the mutant in the absence of 3BNC117, $C_{Ab}$ is the percentage GFP signal of the control virus in the presence of 3BNC117 and $C_{nt}$ is the percentage GFP signal of the control virus in the absence of 3BNC117. For visualization, we enumerated all replicate-combination ratios ($3 \times 3 \times 3 \times 3 = 81$ $R_i$ values per mutant) and plotted their distribution as violin plots. To summarize central tendancy and uncertainty, we performed non-parametric bootstrap resampling. For each bootstrap, we resampled three values with replacement from each $m_{Ab}$, $m_{nt}$, $C_{Ab}$ and $C_{nt}$, computed the mean of each resampled set and calculated $R_i$ from these means. Experiments were performed on two independent days with three biological replicates per condition per day with similar results. AUC calculations were also performed for the 3BNC117-treated growth curves alone with similar results. For transwell experiments, the identical set up was used except a 24-well transwell with a 0.4-μM filter was used (Corning CLS3470). The 293T and MT4 cells were plated across the transwell membrane so there was no direct contact between the cells.

### Crystal structures
PDB files for 5v8m and 6u0l were downloaded from the RCSB protein databank and visualized in ChimeraX v1.10. Epitopes were coloured via custom scripts. For CH607 mutants, the corresponding BG505 residues were first changed to wild-type CH607 residues and the surface electrostatic potential was calculated. The residues were then changed to those recovered from CH607 selection experiments and the electrostatic potential was calculated again.

### Logos plot
The multiple-sequence alignment of all HIV-1 viruses was downloaded from the Los Alamos National Laboratory HIV sequence database (https://www.hiv.lanl.gov/). The columns corresponding to the HxB2 numbering were collected and used to create a LOGOS plot using WebLogo3[61,62].

### Biosafety
All experiments were conducted in a BSL2 laboratory in accordance to biosafety regulations. Institutional protocol approval number 2020-01-010.

### Reporting summary
Further information on research design is available in the Nature Portfolio Reporting Summary linked to this article.

## Data availability
Short-read sequences are available on NCBI under bioproject PRJNA1430132. Source data are provided with this paper. These data consist of datasets used to generate IC$_{50/80}$ values and mutant frequency estimations.

## Code availability
All code used for site scoring is available via CodeOcean at https://doi.org/10.24433/CO.1941144.v2.

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

## Acknowledgements

This study was supported by the National Institutes of Health (NIH) grant U01 AI169385 to M.N., M.C.N., T.H., P.D.B. and M.S.S. NIH grants T32 AI007613 to Cornell Medical School and 5KL2TR001865 to Rockefeller University supported A.C.S., M.C.N. and P.D.B. are Howard

Hughes Medical Institute (HHMI) investigators. This article is subject to HHMI's Open Access to Publications policy. The funders had no role in study design, data collection and analysis, decision to publish or preparation of the manuscript.

## Author contributions

A.C.S., P.D.B. and T.H. conceived and organized the study. A.C.S., P.D.B. and T.H. wrote the paper with input from all authors. A.C.S., S.L. and D.J.P. performed all the selection experiments and all of the pseudoviral screening of putative resistance mutations. A.C.S. and S.L. performed most of the Illumina sequencing library preparations, while V.A.B. performed Illumina sequencing library preparations during assay development. A.C.S. performed all the bioinformatic analysis including phylogenetics and short-read sequencing. A.C.S. conceived of and implemented the scoring heuristic. S.L. performed the cell-to-cell transmission experiments. A.C.S., D.J.P., S.L. and R.P. performed all the cloning of Env expression vectors and single amino acid point mutants. C.L.L. and S.M. produced pseudovirus for and performed TZM-bl neutralization assays of the confirmed resistance mutations on the bnAb panels under the guidance of M.S.S.; and M.C.N., M.C., M.S.S., P.D.B. and T.H. provided key feedback and discussions.

## Competing interests

The following authors declare competing interests: M.C.N. is listed as an inventor on patents for the antibodies 3BNC117 and 10-1074 and M.C. participates in advisory boards for Gilead, Merck and ViiV. The other authors declare no competing interests.

## Additional information

**Extended data** is available for this paper at https://doi.org/10.1038/s41564-026-02347-x.

**Correspondence and requests for materials** should be addressed to Paul D. Bieniasz or Theodora Hatziioannou.

**Extended Data Table 1 | Statistics for comparison of bnAb resistance frequencies prior to selection at different bnAb concentrations**

|  | sum_sq | df | F | PR(>F) |  |  |
|---|---|---|---|---|---|---|
| GroupLabel | 3.817889 | 5 | 6.5763 | 0.000047 |  |  |
| Residual | 8.011627 | 69 | NaN | NaN |  |  |
| Multiple Comparison of Means - Tukey HSD, FWER=0.05 |  |  |  |  |  |  |
| group1 | group2 | meandiff | p-adj | lower | upper | reject |
| ------------------------------------------------------ |  |  |  |  |  |  |
| 100_10-1074 | 10_10-1074 | 0.1118 | 0.9818 | -0.359 | 0.5827 | ##### |
| 100_10-1074 | 1_10-1074 | 0.2592 | 0.5925 | -0.2117 | 0.7301 | ##### |
| 100_3BNC117 | 10_3BNC117 | 0.0626 | 0.9952 | -0.2906 | 0.4157 | ##### |
| 100_3BNC117 | 1_3BNC117 | 0.5704 | 0.0002 | 0.2173 | 0.9236 | TRUE |
| 10_10-1074 | 1_10-1074 | 0.1473 | 0.9408 | -0.3235 | 0.6182 | ##### |
| 10_3BNC117 | 1_3BNC117 | 0.5078 | 0.001 | 0.1547 | 0.861 | TRUE |
| ------------------------------------------------------ |  |  |  |  |  |  |

Tukey's HSD test following one-way ANOVA ($F(5, 69)=6.58$, $p=4.7 \times 10^{-5}$) comparing group means across six antibody conditions (n=15 for 3BNC117 groups, n=9 for 10-1074 groups). Reported values include mean differences, 95% confidence intervals, adjusted $p$-values, and significance calls ($\alpha=0.05$).

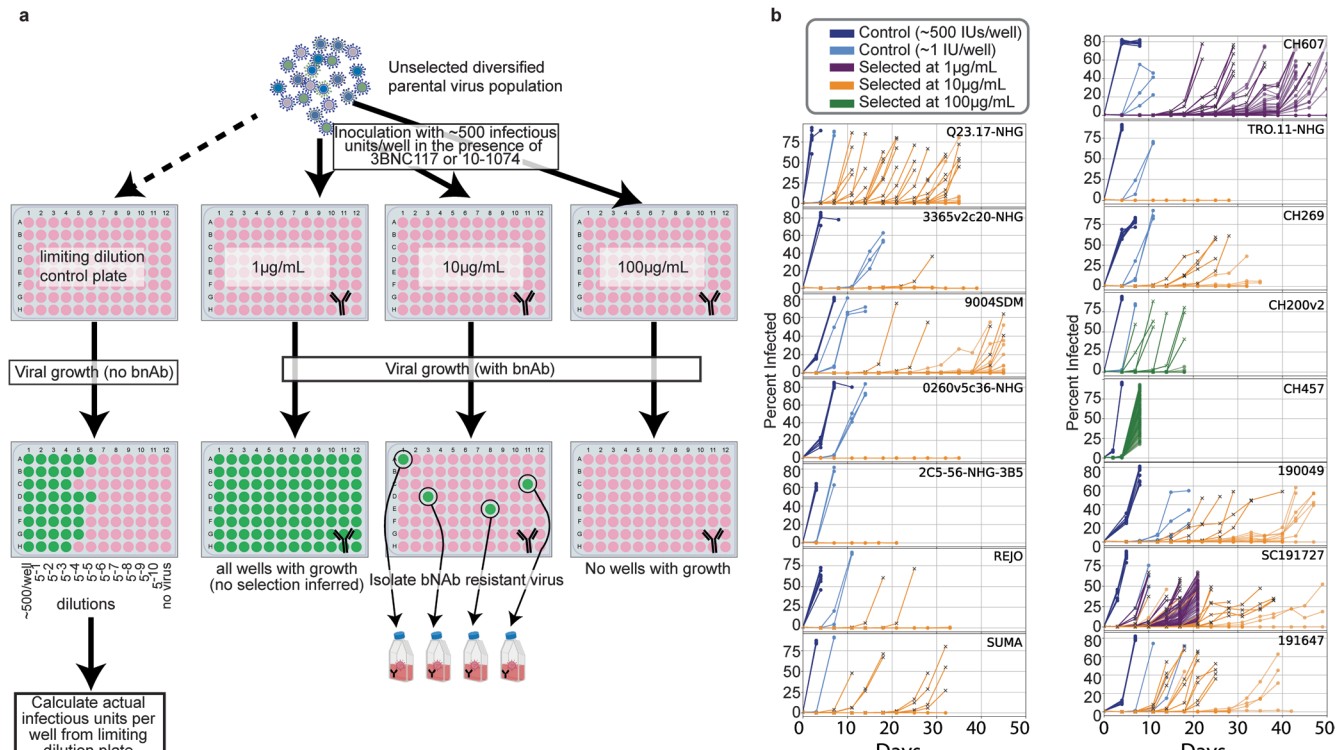

**Extended Data Fig. 1 | Overview of selection experiments and example growth kinetics of 3BNC117 selections. a**, Overview of the selection strategy used in this study. **b**, Data from a subset of selection experiments against the bnAb 3BNC117.

Isolates subsequently sequenced are marked with an x. Dark blue represents the first column in the limiting dilution control plate, light blue is the last column in each row that contained replicating virus in the limiting dilution control plate.

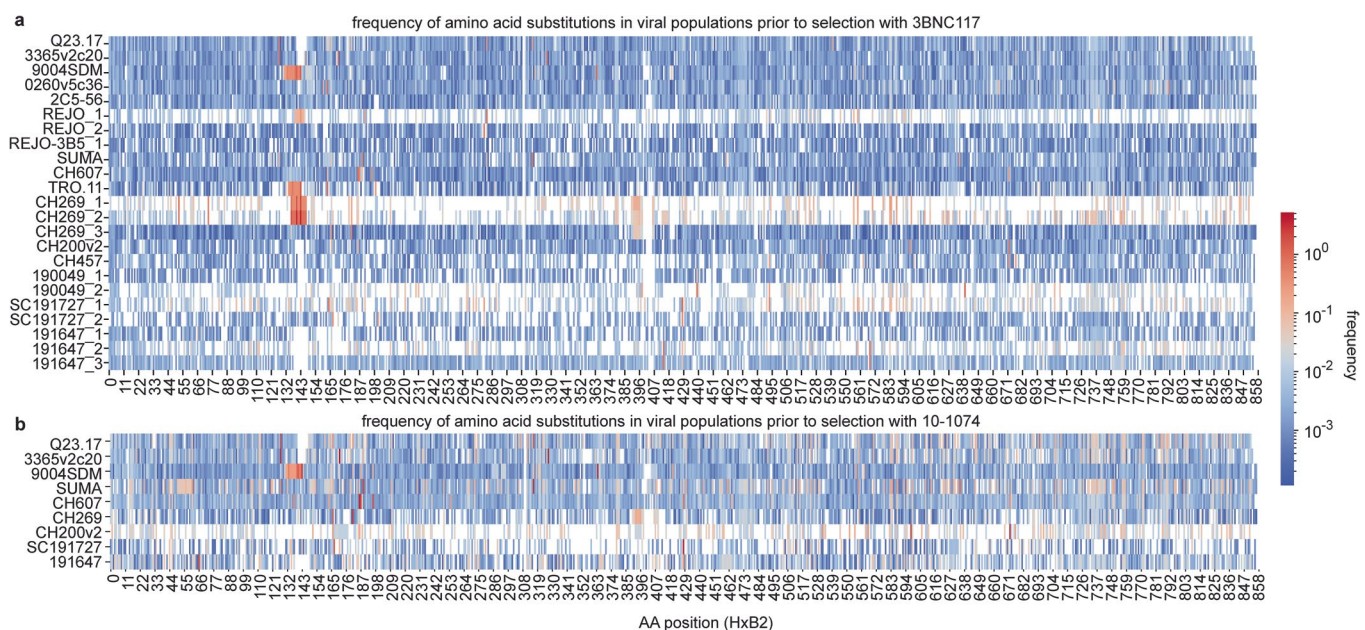

**Extended Data Fig. 2 | Distribution and frequency of variants in pre-selection viral populations. a-b,** Heatmaps showing the frequency of non-synonymous mutations in the viral populations prior to selection for both 3BNC117 (**a**) and 10-1074 (**b**) selections.

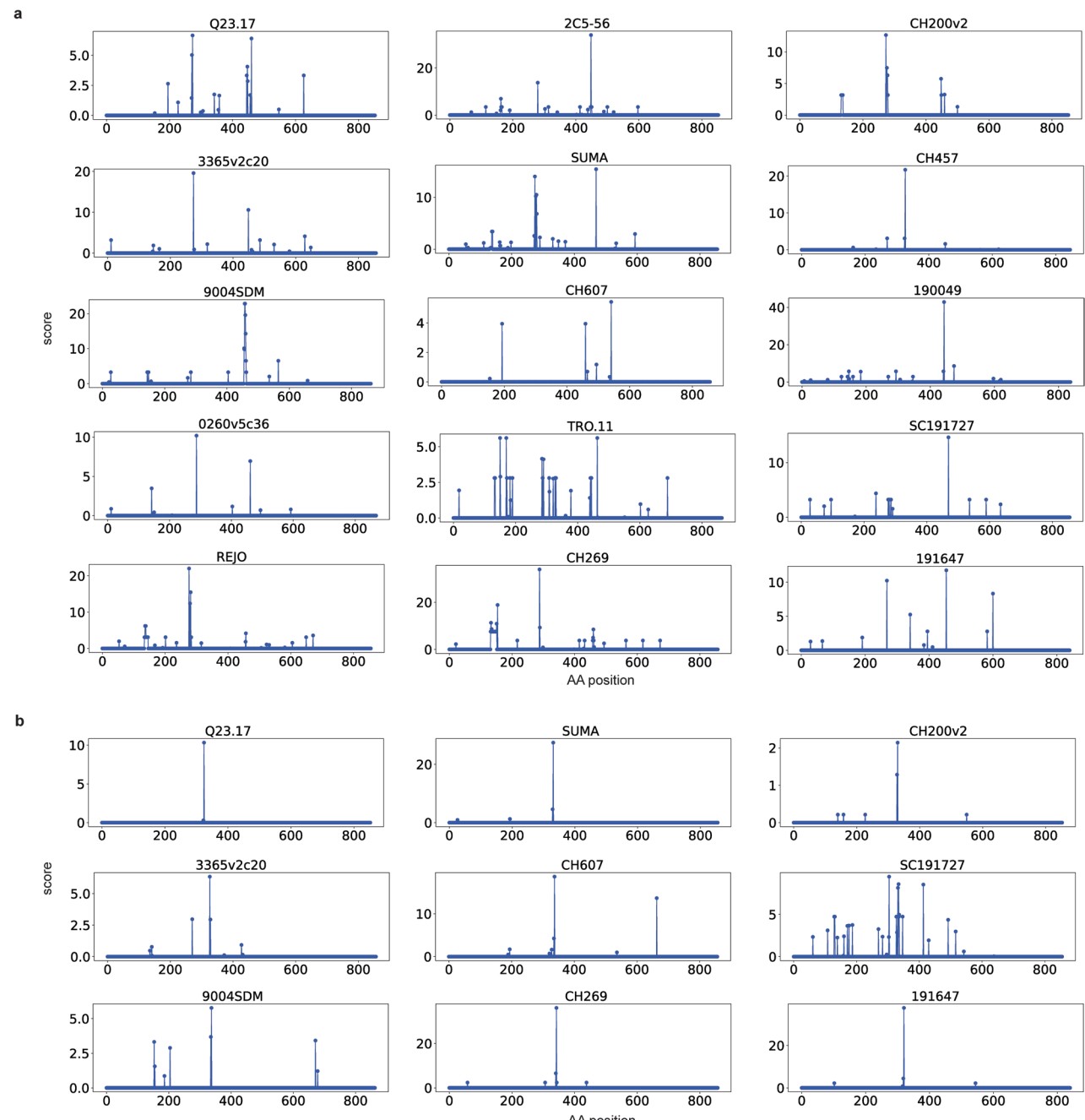

**Extended Data Fig. 3 | Application of scoring heuristic for 3BNC117 and 10-1074 selections. a-b**, Graphs showing the scoring heuristic for putative resistance mutations for each Env is shown for 3BNC117 (**a**) and 10-1074 (**b**). A representative selection experiment was chosen for each viral Env-antibody combination.

**a**

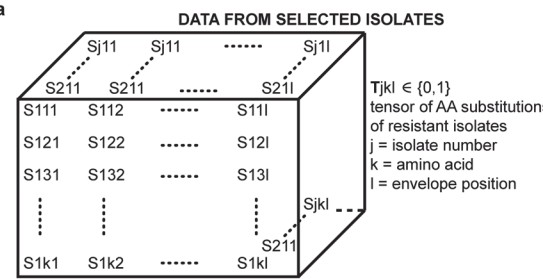

**DATA FROM SELECTED ISOLATES**

$T_{jkl} \in \{0,1\}$
tensor of AA substitutions
of resistant isolates
j = isolate number
k = amino acid
l = envelope position

**VARIANT FREQUENCY IN CONTROL
(UNSELECTED) VIRAL POPULATIONS**

| C11 | C12 | ······ | C1l |
|-----|-----|--------|-----|
| C21 | C22 | ······ | C2l |
| C31 | C32 | ······ | C3l |
| ⋮ | ⋮ | | ⋮ |
| Ck1 | Ck2 | ······ | Ckl |

$M_{kl} \in [0,1]$
Matrix of AA frequencies
in pre-selection viral population
k = amino acid
l = envelope position

**ALGORITHM USED IN PAPER:**

1) Score of amino acid "k" at site "l" $= \sum\limits_{j} t_{jkl} * Log\left(1 - \left(1 - m_{kl}\right)^N\right)$ (N = "N-factor")

2) Take highest scoring fixed site from each sample to identify putative resistance sites

**c**

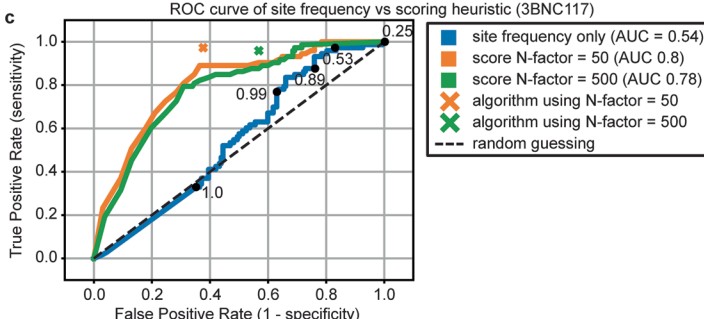

ROC curve of site frequency vs scoring heuristic (3BNC117)

- site frequency only (AUC = 0.54)
- score N-factor = 50 (AUC 0.8)
- score N-factor = 500 (AUC 0.78)
- ✗ algorithm using N-factor = 50
- ✗ algorithm using N-factor = 500
- --- random guessing

**d**

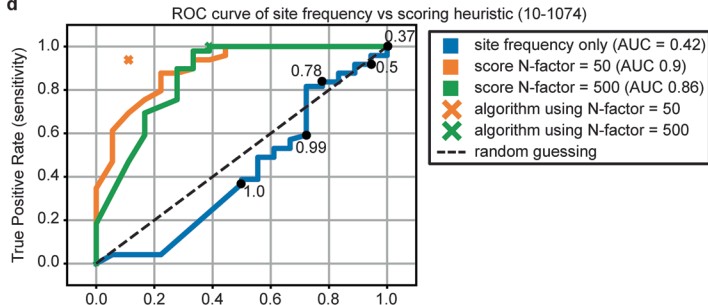

ROC curve of site frequency vs scoring heuristic (10-1074)

- site frequency only (AUC = 0.42)
- score N-factor = 50 (AUC 0.9)
- score N-factor = 500 (AUC 0.86)
- ✗ algorithm using N-factor = 50
- ✗ algorithm using N-factor = 500
- --- random guessing

**b**

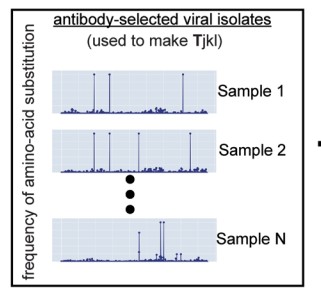

antibody-selected viral isolates
(used to make **T**jkl)

Sample 1
Sample 2
Sample N

**+**

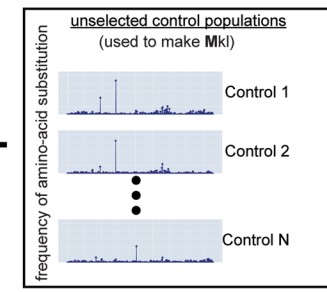

unselected control populations
(used to make **M**kl)

Control 1
Control 2
Control N

**→**

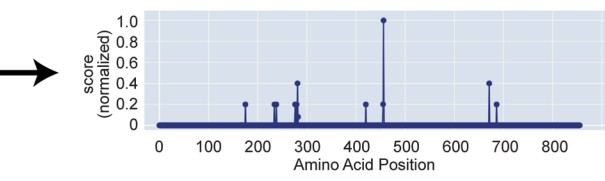

**Extended Data Fig. 4 | Schematic of the resistance scoring system and algorithm and performance compared to traditional methods.** **a-b**, Schematics depicting the scoring function and algorithm used to identify putative resistance mutations in this paper is shown. **c-d**, ROC curves showing the performance of identifying resistance mutations using the algorithm outlined in **a** is shown for 3BNC117 scored sites (**c**) and 10-1074-scored sites (**d**). For each ROC curve, the performance using the rank of sites for each selection experiment based on their score is used to generate the curve, while the "x"s mark the algorithm performance for different N-factors. ROC curves were generated using results from pseudoviral neutralization assays.

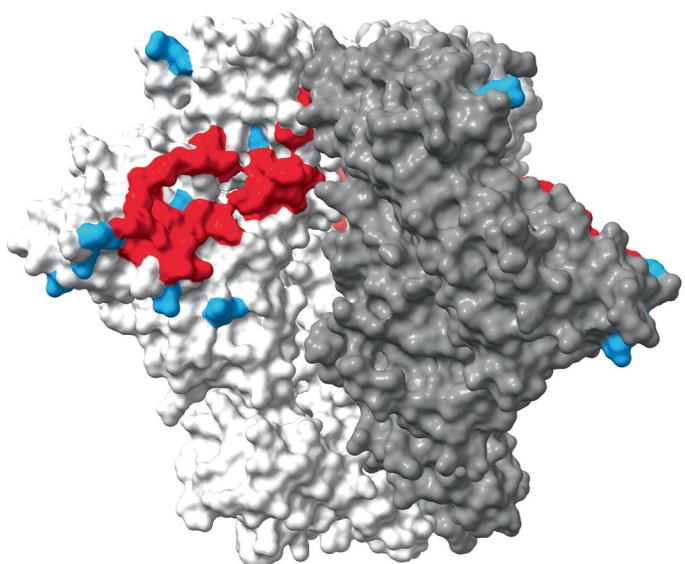

**Extended Data Fig. 5 | 3BNC117 resistance sites outside the CD4 binding site.** Crystal structure of the HIV-1 Env trimer (PDB 5v8m) with the CD4 binding site shown in red and validated 3BNC117 resistance sites identified that lie outside of this region shown in blue.

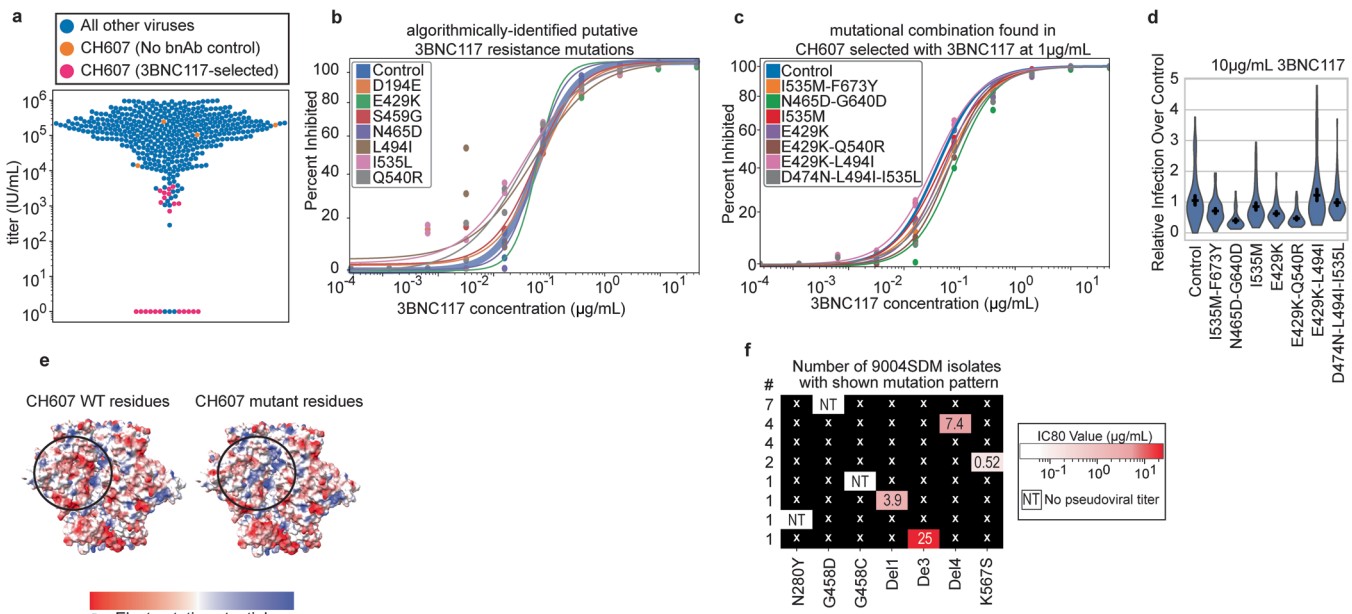

**Extended Data Fig. 6 | Uncommon 3BNC117 escape pathways. a**, Titer of CH607 isolates grown in the absence of 3BNC117 (orange) or selected with 1 μg/mL 3BNC117 (pink) in comparison to all other virus isolates obtained following selection with 3BNC117 (blue). **b-c**, Neutralization curves of viruses pseudotyped with CH607 Env mutants isolated after selection with 3BNC117 at 1 μg/mL. Algorithmically identified putative resistance mutations are show in **b** and combinations found in individual isolates are shown in **c. d**, 293Ts were transfected with an infectious molecular clone of CH607 bearing the indicated mutant and co-cultured with MT4-R5-GFP cells at a 3BNC117 concentration of

10 μg/mL. Relative infection was calculated for each mutant and violin plots show the distribution of relative infection values generated from all replicate combinations ($3^4$ = 81 per mutant, see Methods). Horizontal bars indicate the bootstrap mean and error bars represent the 95% resampling confidence interval.**e**, Electrostatic potential calculated for parental (left) or mutant (right) CH607 Env surface (PDB 5v8m). **f**, Number of 3BNC117 resistant 9004SDM isolates with the indicated selected amino acid substitution. 3BNC117 IC80 values were determined for selected mutants as indicated.

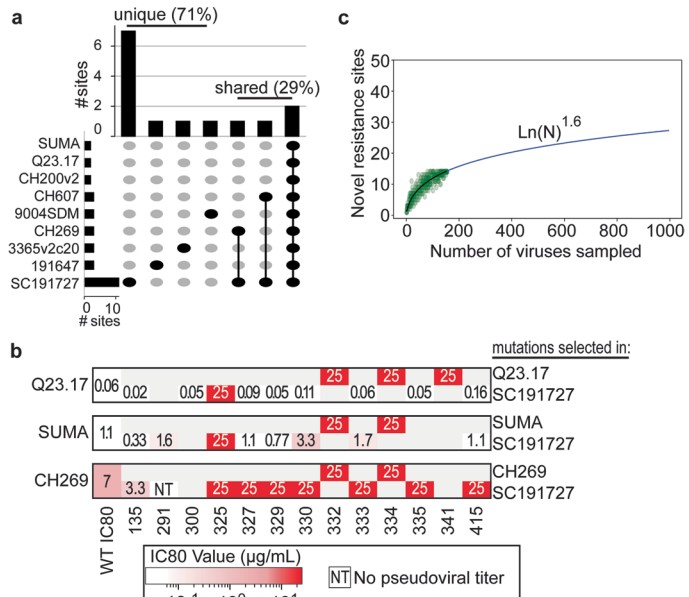

**Extended Data Fig. 7 | Context dependence of unique 10-1074 resistance mutations identified for SC191727. a**, UpSetPlot of 10-1074 resistance sites tested. Each row represents a viral Env, and each column represents a unique combination of resistance sites (dot matrix). The top bar graph shows how many sites fall into each combination, while the side bar graph shows how many resistance sites were identified in each Env. **b**, 10-1074 IC80 values ($\mu$g/mL) for mutations either selected natively (top) or found in the SC191727 backbone and swapped into a heterologous viral Env (bottom). **c**, bootstrap re-sampling of the sequencing data showing the number of novel resistance sites detected for the number of antibody resistance viruses sequenced.

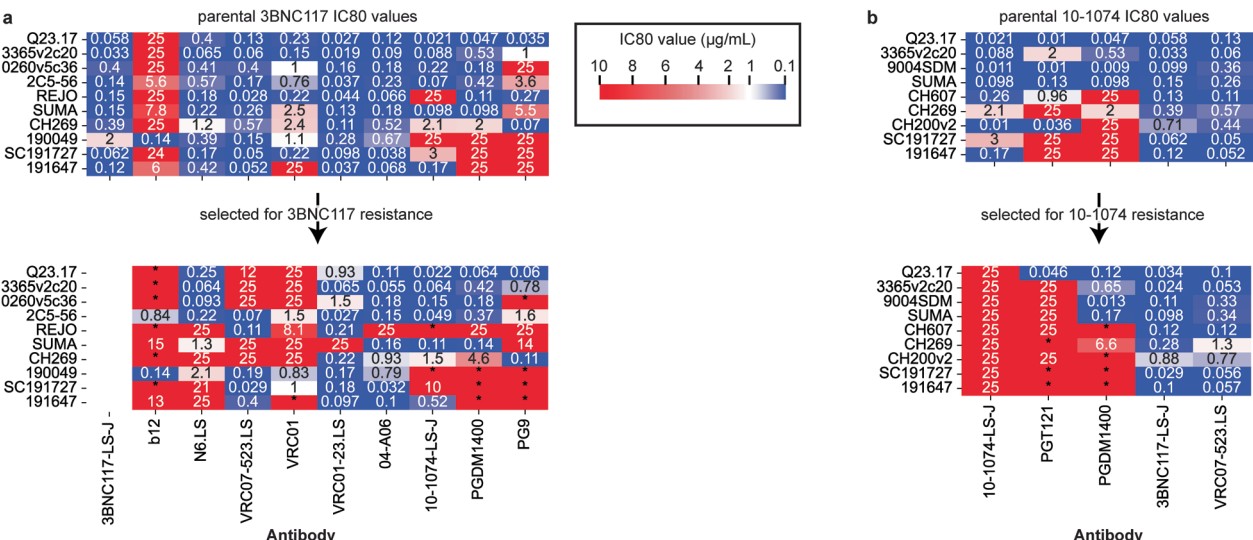

**Extended Data Fig. 8 | 3BNC117 resistance mutations give dual resistance for all bnAbs tested. a**, IC80 values of bnAbs against each parental Env (top) and single amino acid, 3BNC117-resistant, Env mutant that conferred the highest level of resistance to 3BNC117. **b**, Same analysis as in **a** for 10-1074.

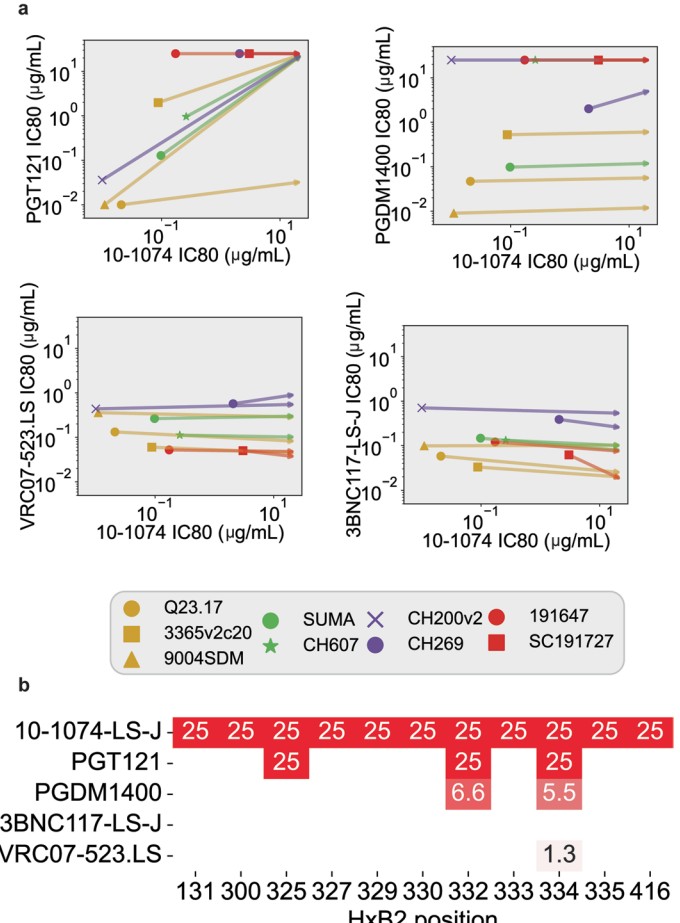

**Extended Data Fig. 9 | Cross resistance of 10-1074 resistance mutations to PGT121 but not other bnAbs. a**, Each circle represents the IC80 values for 10-1074 (x-axis) and a second antibody (y-axis) for each parental Env. Arrows represent the average change in IC80 values due to 10-1074 resistance mutations (geometric mean difference from the parental virus). **b**, Amino-acid positions (HxB2 numbering) where changes confer 10-1074 resistance and cross-resistance to another bnAb indicated.

# Reporting Summary

## Statistics

For all statistical analyses, confirm that the following items are present in the figure legend, table legend, main text, or Methods section.

| n/a | Confirmed | |
|---|---|---|
| ☐ | ☒ | The exact sample size (*n*) for each experimental group/condition, given as a discrete number and unit of measurement |
| ☐ | ☒ | A statement on whether measurements were taken from distinct samples or whether the same sample was measured repeatedly |
| ☐ | ☒ | The statistical test(s) used AND whether they are one- or two-sided *Only common tests should be described solely by name; describe more complex techniques in the Methods section.* |
| ☒ | ☐ | A description of all covariates tested |
| ☒ | ☐ | A description of any assumptions or corrections, such as tests of normality and adjustment for multiple comparisons |
| ☐ | ☒ | A full description of the statistical parameters including central tendency (e.g. means) or other basic estimates (e.g. regression coefficient) AND variation (e.g. standard deviation) or associated estimates of uncertainty (e.g. confidence intervals) |
| ☐ | ☒ | For null hypothesis testing, the test statistic (e.g. *F*, *t*, *r*) with confidence intervals, effect sizes, degrees of freedom and *P* value noted *Give P values as exact values whenever suitable.* |
| ☒ | ☐ | For Bayesian analysis, information on the choice of priors and Markov chain Monte Carlo settings |
| ☒ | ☐ | For hierarchical and complex designs, identification of the appropriate level for tests and full reporting of outcomes |
| ☒ | ☐ | Estimates of effect sizes (e.g. Cohen's *d*, Pearson's *r*), indicating how they were calculated |

*Our web collection on statistics for biologists contains articles on many of the points above.*

## Software and code

Policy information about availability of computer code

| Data collection | Attune Cytometric Software v6.2.1 |
|---|---|
| Data analysis | Geneious Prime 2025.2.2<br>RaxML v4.0<br>bwa Version: 0.7.17-r1188<br>LoFreq Version 3<br>VirVarSeq Version 1<br>ChimeraX-1.10<br>Python v3.12.3 |

For manuscripts utilizing custom algorithms or software that are central to the research but not yet described in published literature, software must be made available to editors and reviewers. We strongly encourage code deposition in a community repository (e.g. GitHub). See the Nature Portfolio guidelines for submitting code & software for further information.

## Data

Policy information about [availability of data](availability of data)
All manuscripts must include a [data availability statement](data availability statement). This statement should provide the following information, where applicable:

- Accession codes, unique identifiers, or web links for publicly available datasets
- A description of any restrictions on data availability
- For clinical datasets or third party data, please ensure that the statement adheres to our [policy](policy)

> Short read sequences is available on NCBI under bioproject PRJNA1430132. Raw data from which all graphs were derived has been uploaded to Figshare

## Research involving human participants, their data, or biological material

Policy information about studies with [human participants or human data](human participants or human data). See also policy information about [sex, gender (identity/presentation), and sexual orientation](sex, gender (identity/presentation), and sexual orientation) and [race, ethnicity and racism](race, ethnicity and racism).

| | |
|---|---|
| Reporting on sex and gender | *Use the terms sex (biological attribute) and gender (shaped by social and cultural circumstances) carefully in order to avoid confusing both terms. Indicate if findings apply to only one sex or gender; describe whether sex and gender were considered in study design; whether sex and/or gender was determined based on self-reporting or assigned and methods used.*<br>*Provide in the source data disaggregated sex and gender data, where this information has been collected, and if consent has been obtained for sharing of individual-level data; provide overall numbers in this Reporting Summary. Please state if this information has not been collected.*<br>*Report sex- and gender-based analyses where performed, justify reasons for lack of sex- and gender-based analysis.* |
| Reporting on race, ethnicity, or other socially relevant groupings | *Please specify the socially constructed or socially relevant categorization variable(s) used in your manuscript and explain why they were used. Please note that such variables should not be used as proxies for other socially constructed/relevant variables (for example, race or ethnicity should not be used as a proxy for socioeconomic status).*<br>*Provide clear definitions of the relevant terms used, how they were provided (by the participants/respondents, the researchers, or third parties), and the method(s) used to classify people into the different categories (e.g. self-report, census or administrative data, social media data, etc.)*<br>*Please provide details about how you controlled for confounding variables in your analyses.* |
| Population characteristics | *Describe the covariate-relevant population characteristics of the human research participants (e.g. age, genotypic information, past and current diagnosis and treatment categories). If you filled out the behavioural & social sciences study design questions and have nothing to add here, write "See above."* |
| Recruitment | *Describe how participants were recruited. Outline any potential self-selection bias or other biases that may be present and how these are likely to impact results.* |
| Ethics oversight | *Identify the organization(s) that approved the study protocol.* |

Note that full information on the approval of the study protocol must also be provided in the manuscript.

# Field-specific reporting

Please select the one below that is the best fit for your research. If you are not sure, read the appropriate sections before making your selection.

☒ Life sciences   ☐ Behavioural & social sciences   ☐ Ecological, evolutionary & environmental sciences

For a reference copy of the document with all sections, see [nature.com/documents/nr-reporting-summary-flat.pdf](nature.com/documents/nr-reporting-summary-flat.pdf)

# Life sciences study design

All studies must disclose on these points even when the disclosure is negative.

| | |
|---|---|
| Sample size | We aimed to isolate at least 5 resistant viral isolates per parental strain. If more were selected, we would collect additional samples. All experiments were performed with at least 3 biological replicates on 2 independent days |
| Data exclusions | isolates were excluded from analysis if they had low read coverage or poor quality |
| Replication | Multiple replicates were performed in parallel for each selection experiment. All neutralization assays were performed at least twice to confirm IC80 values. All resistant variants were tested independently in a second laboratory to confirm resistance. Cell-to-cell transmission experiments were performed in 3 biological replicates on two independent days. For selection experiments, some viruses underwent the entire experimental procedure more than once with the same sites detected in independent experiments. |
| Randomization | N/A |
| Blinding | Blinding was not performed because condition identity was required for experimental setup and for prespecified normalization/QC, and with |

| Blinding | a small team separate blinded analysis was not operationally feasible; bias was minimized by applying uniform gating and automated analysis pipelines |
|---|---|

# Reporting for specific materials, systems and methods

We require information from authors about some types of materials, experimental systems and methods used in many studies. Here, indicate whether each material, system or method listed is relevant to your study. If you are not sure if a list item applies to your research, read the appropriate section before selecting a response.

## Materials & experimental systems

| n/a | Involved in the study |
|---|---|
| ☐ | ☒ Antibodies |
| ☐ | ☒ Eukaryotic cell lines |
| ☒ | ☐ Palaeontology and archaeology |
| ☒ | ☐ Animals and other organisms |
| ☒ | ☐ Clinical data |
| ☒ | ☐ Dual use research of concern |
| ☒ | ☐ Plants |

## Methods

| n/a | Involved in the study |
|---|---|
| ☒ | ☐ ChIP-seq |
| ☐ | ☒ Flow cytometry |
| ☒ | ☐ MRI-based neuroimaging |

## Antibodies

| Antibodies used | 3BNC117.LS & 10-1074.LS (provided by Michel Nussensweig) |
|---|---|
| Validation | All antibodies were tested for neutralization potency upon receipt against a panel of known susceptible and resistant viruses |

## Eukaryotic cell lines

Policy information about cell lines and Sex and Gender in Research

| Cell line source(s) | MT4.R5.GFP and MT4.R5.GFP-CD4high were derived from MT4 cells (aids reagent program ARP-120, now through BEI) 293T (ATCC CRL-3216) |
|---|---|
| Authentication | None of the cell lines were authenticated |
| Mycoplasma contamination | All cell lines were tested for mycoplasma contamination and were found to not be contaminated |
| Commonly misidentified lines (See ICLAC register) | *Name any commonly misidentified cell lines used in the study and provide a rationale for their use.* |

## Plants

| Seed stocks | *Report on the source of all seed stocks or other plant material used. If applicable, state the seed stock centre and catalogue number. If plant specimens were collected from the field, describe the collection location, date and sampling procedures.* |
|---|---|
| Novel plant genotypes | *Describe the methods by which all novel plant genotypes were produced. This includes those generated by transgenic approaches, gene editing, chemical/radiation-based mutagenesis and hybridization. For transgenic lines, describe the transformation method, the number of independent lines analyzed and the generation upon which experiments were performed. For gene-edited lines, describe the editor used, the endogenous sequence targeted for editing, the targeting guide RNA sequence (if applicable) and how the editor was applied.* |
| Authentication | *Describe any authentication procedures for each seed stock used or novel genotype generated. Describe any experiments used to assess the effect of a mutation and, where applicable, how potential secondary effects (e.g. second site T-DNA insertions, mosiacism, off-target gene editing) were examined.* |

## Flow Cytometry

### Plots

Confirm that:

☒ The axis labels state the marker and fluorochrome used (e.g. CD4-FITC).

☒ The axis scales are clearly visible. Include numbers along axes only for bottom left plot of group (a 'group' is an analysis of identical markers).

☒ All plots are contour plots with outliers or pseudocolor plots.

☒ A numerical value for number of cells or percentage (with statistics) is provided.

## Methodology

Sample preparation
> all cells were fixed in 2% PFA for 5 minutes at room temperature. The cells were pelleted at 500xg and resuspended in PBS for analysus

Instrument
> Attune NxT

Software
> Attune Cytometric Software

Cell population abundance
> all samples contained >40% target cells after exclusion of doublets and debris

Gating strategy
> 1) FSC-A/SSC-A (main cell population)
> 2) SSC-A/SSC-H
> 3) GFP+ (negative set so no more that 0.1% of the population was GFP+ in the negative control)

☒ Tick this box to confirm that a figure exemplifying the gating strategy is provided in the Supplementary Information.

