## [Peer Review File · Nature Microbiology]

Diverse paths to broadly neutralizing antibody escape among HIV-1 strains.

Corresponding Author: Professor Theodora Hatzioannou

Version 0:

Reviewer comments:

Reviewer #1

(Remarks to the Author)

I was Referee #1 on the original submission of this manuscript to Nature so I will only make a few summary remarks and comments on the authors' response.

The original paper described a well conducted study that I considered fell a little below the level of impact to make it into Nature. However, it is an important study and well worthy of publication.

Two comments on the authors' response:

1. The authors did not respond to my comment about the translational studies being alluded to but not clearly explained. Here, I understood that the authors envisaged translation: "These data provide a rationale for selecting bnAb combinations that are most likely to achieve treatment success" and "Nevertheless, approaches such as those documented here in can identify combinations of bnAbs with a higher barriers to escape that could guide strategies to minimize the development of resistance". I was looking for the authors, given their experiences in this study, to provide a roadmap for the field about how this might be achieved. Which viruses to use, how many, what conditions etc etc.

2. For the structural comment, I was interested to see mapped on a (gp120?) structure where the significant mutations distant from the CD4 binding site were. I did not really get that from the two figures described.

Again, a fine manuscript-the above are suggestions for improvement.

(Remarks on code availability)

Reviewer #2

(Remarks to the Author)

This is a well-done and timely manuscript delving into escape pathways from bNabs 3BNC117 and 10-1074. These data are highly clinically relevant. The authors addressed all the reviewers comments.

I have minor comments.

In response to Reviewer 2's point about Line 195, I wonder if the reviewer meant regions of the mutations (ie, did almost all resistance mutations arise in Loop D or V5 for 3BNC117 and glycan 332 for 10-1074 as it looks like they might from Figure 2a and b?)

On Line 240, 10-1074 and PGT121 not only target the same glycan epitope, they are clonal relatives, inferred to have derived from the same ancestral naive B cell (I think?).

(Remarks on code availability)

I have no expertise to assess code.

Reviewer #3

(Remarks to the Author)

Summary

This manuscript by Stabell et al. investigates HIV-1 escape mutations from the broadly neutralizing antibodies 10-1074 and

3BNC117 using a replication-competent viral system. While escape mutations for these bnAbs were recently mapped by Radford et al (JV, 2025). using two divergent Env strains, the present study extends this work by analyzing resistance across 15 primary HIV-1 isolates and assessing cross-resistance against an additional panel of nine bnAbs. This expanded scope provides a broader experimental basis for understanding resistance mechanisms and for informing bnAb combination strategies.

The experimental and analytic approaches are well thought out. In particular, resistance mutations arise through natural Env diversification during multi-cycle replication and are selected in a 96-well bottlenecked format, where resistant viruses typically expand from a single dominant lineage. This strategy can reduce several technical confounders associated with bulk selection or single-round-infection of mutant libraries. The resistance scoring framework, which integrates unselected mutation frequency with recurrence across independent resistant isolates, is reasonable and conceptually sound.

Overall, this is a strong and complementary study with clear relevance to bnAb therapeutic design, although several aspects of contextualization and data presentation could be improved.

Major Concerns

1. The manuscript does not adequately discuss recent work by Radford et al. demonstrating strain-dependent escape pathways for 10-1074 and 3BNC117.

- The statement (line 55) that it is "unknown whether HIV-1 isolates differ in their ability to escape bnAb neutralization" is inaccurate and should be revised.

- A clearer comparison explaining how this study extends or complements prior escape-mapping work would strengthen the framing and help avoid overstatement of novelty.

2. Resistance mutation mapping and sequencing quality remain unclear.

- The authors used Nextera XT-based Illumina sequencing to profile mutations in ~2.5 kb env amplicons, but sequencing quality metrics (e.g., read depth and coverage) were not provided. Given that this approach can produce uneven coverage, conclusions regarding single- versus multi-site resistance mutations are difficult to evaluate technically without additional sequencing QC data; "not observed" may reflect true biological absence of mutations but may also result from lack of detection.

- The resistance scoring framework incorporates mutation frequencies from the "pre-selection population," but this term is ambiguous. It is unclear whether it refers to virus populations at an early stage of the experiment (e.g., virus produced from transfected cells or after the first passage) or to viruses from the control plates that underwent several passages without selection. If the latter (which likely the case), a clearer term (e.g., "no-selection population") would improve clarity.

- In addition, it is unclear how the control plates were handled prior to sequencing (e.g., number of passages, duration of culture).

- While the scoring framework is reasonable, the manuscript would benefit from a clearer discussion of its assumptions, limitations and significance, particularly in cases where multiple mutations are present; and in comparison to other approaches using bulk /single-infection selection of mutant libraries.

Minor Concerns

- Clarify the frequency thresholds used to define dominant resistance mutations within individual wells.

- Indicate how often multiple resistance mutations were observed relative to single-mutation outcomes.

- Specify whether any regions of env were excluded from analysis due to poor coverage or alignment ambiguity.

- Consider adding a schematic summarizing the experimental workflow of the resistance scoring system (including the role of the control plates) to improve clarity.

(Remarks on code availability)

Decision Letter:

7th January 2026

Dear Theodora,

Thank you for your patience while your manuscript "Diverse paths to broadly neutralizing antibody escape among HIV-1 strains." was under peer-review at Nature Microbiology. It has now been seen by 3 referees, whose expertise and comments you will find at the of this email. You will see from their comments below that while they find your work of interest, some important points are raised. We are very interested in the possibility of publishing your study in Nature Microbiology, but would like to consider your response to these concerns in the form of a revised manuscript before we make a final decision on publication.

In particular, you will see that referee #1 requests a revised discussion including an outline on how improved combinations of bnAbs can be identified. Referee #3 asks for more clarifications on the quality of the data, as well as to discuss your work with recent literature. The rest referees' reports are clear and the remaining issues should be straightforward to address.

If you have not done so already please begin to revise your manuscript so that it conforms to our Article format instructions at <http://www.nature.com/nmicrobiol/info/final-submission/>

The usual length limit for a Nature Microbiology Article is six display items (figures or tables) and 3,000 words. We have some flexibility, and can allow a revised manuscript at 3,500 words, but please consider this a firm upper limit. There is a trade-off of ~250 words per display item, so if you need more space, you could move a Figure or Table to Supplementary Information.

Some reduction could be achieved by focusing any introductory material and moving it to the start of your opening 'bold' paragraph, whose function is to outline the background to your work, describe in a sentence your new observations, and explain your main conclusions. The discussion should also be limited. Methods should be described in a separate section following the discussion, we do not place a word limit on Methods.

Nature Microbiology titles should give a sense of the main new findings of a manuscript, and should not contain punctuation. Please keep in mind that we strongly discourage active verbs in titles, and that they should ideally fit within 90 characters each (including spaces).

Please include a data availability statement as a separate section after Methods but before references, under the heading "Data Availability". This section should inform readers about the availability of the data used to support the conclusions of your study. This information includes accession codes to public repositories (data banks for protein, DNA or RNA sequences, microarray, proteomics data etc...), references to source data published alongside the paper, unique identifiers such as URLs to data repository entries, or data set DOIs, and any other statement about data availability. At a minimum, you should include the following statement: "The data that support the findings of this study are available from the corresponding author upon request", mentioning any restrictions on availability. If DOIs are provided, we also strongly encourage including these in the Reference list (authors, title, publisher (repository name), identifier, year). For more guidance on how to write this section please see: <http://www.nature.com/authors/policies/data/data-availability-statements-data-citations.pdf>

To improve the accessibility of your paper to readers from other research areas, please pay particular attention to the wording of the paper's opening bold paragraph, which serves both as an introduction and as a brief, non-technical summary in about 150 words. If, however, you require one or two extra sentences to explain your work clearly, please include them even if the paragraph is over-length as a result. The opening paragraph should not contain references. Because scientists from other sub-disciplines will be interested in your results and their implications, it is important to explain essential but specialised terms concisely. We suggest you show your summary paragraph to colleagues in other fields to uncover any problematic concepts.

If your paper is accepted for publication, we will edit your display items electronically so they conform to our house style and will reproduce clearly in print. If necessary, we will re-size figures to fit single or double column width. If your figures contain several parts, the parts should form a neat rectangle when assembled. Choosing the right electronic format at this stage will speed up the processing of your paper and give the best possible results in print. We would like the figures to be supplied as vector files - EPS, PDF, AI or postscript (PS) file formats (not raster or bitmap files), preferably generated with vector-graphics software (Adobe Illustrator for example). Please try to ensure that all figures are non-flattened and fully editable. All images should be at least 300 dpi resolution (when figures are scaled to approximately the size that they are to be printed at) and in RGB colour format. Please do not submit Jpeg or flattened TIFF files. Please see also 'Guidelines for Electronic Submission of Figures' at the end of this letter for further detail.

Figure legends must provide a brief description of the figure and the symbols used, within 350 words, including definitions of any error bars employed in the figures.

When submitting the revised version of your manuscript, please pay close attention to our [href="https://www.nature.com/nature-research/editorial-policies/image-integrity">Digital Image Integrity Guidelines.](https://www.nature.com/nature-research/editorial-policies/image-integrity) and to the following points below:

EXTENDED DATA FIGURES

Please include a statement before the acknowledgements naming the author to whom correspondence and requests for materials should be addressed.

Finally, we require authors to include a statement of their individual contributions to the paper -- such as experimental work, project planning, data analysis, etc. -- immediately after the acknowledgements. The statement should be short, and refer to authors by their initials. For details please see the Authorship section of our joint Editorial policies at http://www.nature.com/authors/editorial_policies/authorship.html

* include a point-by-point response to any editorial suggestions and to our referees. Please include your response to the editorial suggestions in your cover letter, and please upload your response to the referees as a separate document.

* ensure it complies with our format requirements for Letters as set out in our guide to authors at www.nature.com/nmicrobiol/info/gta/

* state in a cover note the length of the text, methods and legends; the number of references; number and estimated final size of figures and tables

* resubmit electronically if possible using the link below to access your home page:

Link Redacted

*This url links to your confidential homepage and associated information about manuscripts you may have submitted or be reviewing for us. If you wish to forward this e-mail to co-authors, please delete this link to your homepage first.

Please ensure that all correspondence is marked with your Nature Microbiology reference number in the subject line.

Nature Microbiology is committed to improving transparency in authorship. As part of our efforts in this direction, we are now requesting that all authors identified as 'corresponding author' on published papers create and link their Open Researcher and Contributor Identifier (ORCID) with their account on the Manuscript Tracking System (MTS), prior to acceptance. This applies to primary research papers only. ORCID helps the scientific community achieve unambiguous attribution of all scholarly contributions. You can create and link your ORCID from the home page of the MTS by clicking on 'Modify my Springer Nature account'. For more information please visit www.springernature.com/orcid.

We hope to receive your revised paper within three weeks. If you cannot send it within this time, please let us know.

Yours sincerely,

Reviewer Expertise:

Referee #1: HIV, antibodies

Referee #2: HIV, antibodies

Referee #3: HIV, sequencing

Reviewers Comments:

Reviewer #1 (Remarks to the Author):

I was Referee #1 on the original submission of this manuscript to Nature so I will only make a few summary remarks and comments on the authors' response.

The original paper described a well conducted study that I considered fell a little below the level of impact to make it into Nature. However, it is an important study and well worthy of publication.

Two comments on the authors' response:

1. The authors did not respond to my comment about the translational studies being alluded to but not clearly explained. Here, I understood that the authors envisaged translation: "These data provide a rationale for selecting bnAb combinations that are most likely to achieve treatment success" and "Nevertheless, approaches such as those documented here in can identify combinations of bnAbs with a higher barriers to escape that could guide strategies to minimize the development of resistance". I was looking for the authors, given their experiences in this study, to provide a roadmap for the field about how this might be achieved. Which viruses to use, how many, what conditions etc etc.

2. For the structural comment, I was interested to see mapped on a (gp120?) structure where the significant mutations distant from the CD4 binding site were. I did not really get that from the two figures described.

Again, a fine manuscript—the above are suggestions for improvement.

Reviewer #2 (Remarks to the Author):

This is a well-done and timely manuscript delving into escape pathways from bnAbs 3BNC117 and 10-1074. These data are highly clinically relevant. The authors addressed all the reviewers comments.

I have minor comments.

In response to Reviewer 2's point about Line 195, I wonder if the reviewer meant regions of the mutations (ie, did almost all resistance mutations arise in Loop D or V5 for 3BNC117 and glycan 332 for 10-1074 as it looks like they might from Figure 2a and b?)

On Line 240, 10-1074 and PGT121 not only target the same glycan epitope, they are clonal relatives, inferred to have derived from the same ancestral naive B cell (I think?).

Reviewer #2 (Remarks on code availability):

I have no expertise to assess code.

Reviewer #3 (Remarks to the Author):

Summary

This manuscript by Stabell et al. investigates HIV-1 escape mutations from the broadly neutralizing antibodies 10-1074 and 3BNC117 using a replication-competent viral system. While escape mutations for these bnAbs were recently mapped by Radford et al (JV, 2025). using two divergent Env strains, the present study extends this work by analyzing resistance across 15 primary HIV-1 isolates and assessing cross-resistance against an additional panel of nine bnAbs. This expanded scope provides a broader experimental basis for understanding resistance mechanisms and for informing bnAb combination strategies.

The experimental and analytic approaches are well thought out. In particular, resistance mutations arise through natural Env diversification during multi-cycle replication and are selected in a 96-well bottlenecked format, where resistant viruses typically expand from a single dominant lineage. This strategy can reduce several technical confounders associated with bulk selection or single-round-infection of mutant libraries. The resistance scoring framework, which integrates unselected mutation frequency with recurrence across independent resistant isolates, is reasonable and conceptually sound.

Overall, this is a strong and complementary study with clear relevance to bnAb therapeutic design, although several aspects of contextualization and data presentation could be improved.

Major Concerns

1. The manuscript does not adequately discuss recent work by Radford et al. demonstrating strain-dependent escape pathways for 10-1074 and 3BNC117.

- The statement (line 55) that it is “unknown whether HIV-1 isolates differ in their ability to escape bnAb neutralization” is inaccurate and should be revised.

- A clearer comparison explaining how this study extends or complements prior escape-mapping work would strengthen the framing and help avoid overstatement of novelty.

2. Resistance mutation mapping and sequencing quality remain unclear.

- The authors used Nextera XT–based Illumina sequencing to profile mutations in ~2.5 kb env amplicons, but sequencing quality metrics (e.g., read depth and coverage) were not provided. Given that this approach can produce uneven coverage, conclusions regarding single- versus multi-site resistance mutations are difficult to evaluate technically without additional sequencing QC data; “not observed” may reflect true biological absence of mutations but may also result from lack of detection.

- The resistance scoring framework incorporates mutation frequencies from the “pre-selection population,” but this term is ambiguous. It is unclear whether it refers to virus populations at an early stage of the experiment (e.g., virus produced from transfected cells or after the first passage) or to viruses from the control plates that underwent several passages without selection. If the latter (which likely the case), a clearer term (e.g., “no-selection population”) would improve clarity.

- In addition, it is unclear how the control plates were handled prior to sequencing (e.g., number of passages, duration of culture).

- While the scoring framework is reasonable, the manuscript would benefit from a clearer discussion of its assumptions, limitations and significance, particularly in cases where multiple mutations are present; and in comparison to other approaches using bulk /single-infection selection of mutant libraries.

Minor Concerns

- Clarify the frequency thresholds used to define dominant resistance mutations within individual wells.

- Indicate how often multiple resistance mutations were observed relative to single-mutation outcomes.

- Specify whether any regions of env were excluded from analysis due to poor coverage or alignment ambiguity.
- Consider adding a schematic summarizing the experimental workflow of the resistance scoring system (including the role of the control plates) to improve clarity.

Version 1:

Reviewer comments:

Reviewer #3

(Remarks to the Author)

Overall, the revised version addresses the main technical and contextual concerns raised in the previous review.

The authors have revised the previous statement “It is unknown whether HIV-1 isolates differ in their ability to escape bnAb neutralization” to “Evaluation of bnAb escape across a panel of HIV-1 primary isolates that span global viral diversity has not been previously attempted, and it is unknown whether there is a subtype/geographic variation in the propensity of HIV-1 to acquire bnAb resistance,” thereby emphasizing global testing across a panel of HIV-1 primary isolates in this study, and have added a clearer discussion comparing their approach with prior deep mutational scanning studies. The revised manuscript text more appropriately contextualizes novelty and scope, and I consider this issue adequately addressed.

Concerns regarding sequencing quality and resistance mutation mapping have been clarified through added methodological detail, including explicit variant frequency thresholds, reporting of average read depth, handling of control populations, and justification for focusing on dominant variants arising from bottlenecked selections. Given the experimental design, these clarifications render the conclusions technically reasonable.

Data transparency: The revised manuscript reports only the average read depth of 930 reads/nucleotide for the 2.5-kb env amplicon. Public deposition of the raw (or minimally processed) sequencing data in a standard repository—as a reference dataset for future studies—would allow interested readers to reproduce the analysis and independently evaluate read-depth distribution and coverage uniformity. I encourage the authors to make these data publicly accessible to strengthen transparency and reproducibility.

Ambiguities surrounding the definition and handling of control populations are now resolved, and the added discussion of the resistance-scoring framework and its assumptions strengthens the manuscript.

All minor points have been satisfactorily addressed.

(Remarks on code availability)

Decision Letter:

Our ref: NMICROBIOL-25114290A

23rd February 2026

Dear Theodora,

Thank you for submitting your revised manuscript "Diverse paths to broadly neutralizing antibody escape among HIV-1 strains." (NMICROBIOL-25114290A) and sorry for the delay in our response. Nonetheless, it has now been seen by the original referees and their comments are below. The reviewers find that the paper has improved in revision, and therefore we'll be happy in principle to publish it in Nature Microbiology, pending minor revisions to satisfy the referees' final requests and to comply with our editorial and formatting guidelines.

Thank you again for your interest in Nature Microbiology Please do not hesitate to contact me if you have any questions.

Sincerely,

Reviewer #3 (Remarks to the Author):

Overall, the revised version addresses the main technical and contextual concerns raised in the previous review.

The authors have revised the previous statement "It is unknown whether HIV-1 isolates differ in their ability to escape bnAb neutralization" to "Evaluation of bnAb escape across a panel of HIV-1 primary isolates that span global viral diversity has not been previously attempted, and it is unknown whether there is a subtype/geographic variation in the propensity of HIV-1 to acquire bnAb resistance," thereby emphasizing global testing across a panel of HIV-1 primary isolates in this study, and have added a clearer discussion comparing their approach with prior deep mutational scanning studies. The revised manuscript text more appropriately contextualizes novelty and scope, and I consider this issue adequately addressed.

Concerns regarding sequencing quality and resistance mutation mapping have been clarified through added methodological detail, including explicit variant frequency thresholds, reporting of average read depth, handling of control populations, and justification for focusing on dominant variants arising from bottlenecked selections. Given the experimental design, these clarifications render the conclusions technically reasonable.

Data transparency: The revised manuscript reports only the average read depth of 930 reads/nucleotide for the 2.5-kb env amplicon. Public deposition of the raw (or minimally processed) sequencing data in a standard repository—as a reference dataset for future studies—would allow interested readers to reproduce the analysis and independently evaluate read-depth distribution and coverage uniformity. I encourage the authors to make these data publicly accessible to strengthen transparency and reproducibility.

Ambiguities surrounding the definition and handling of control populations are now resolved, and the added discussion of the resistance-scoring framework and its assumptions strengthens the manuscript.

All minor points have been satisfactorily addressed.

Version 2:

Decision Letter:

30th March 2026

Dear Professor Hatzioannou,

I am pleased to accept your Article "Diverse paths to broadly neutralizing antibody escape among HIV-1 strains." for publication in Nature Microbiology. Thank you for having chosen to submit your work to us and many congratulations.

Authors may need to take specific actions to achieve compliance with funder and institutional open access mandates. If your research is supported by a funder that requires immediate open access (e.g. according to [Plan S principles](https://www.springernature.com/gp/open-science/plan-s-compliance) or the [NIH public access policy](https://www.springernature.com/gp/open-science/us-federal-agency-compliance)) then you should select the gold OA route, and we will direct you to the compliant route where possible. Because authors warrant under our subscription licensing terms that they haven't committed to licensing any version of their article under a licence inconsistent with the terms of our agreement – including the applicable embargo period – publication under the subscription model isn't suitable for authors whose funders require no embargo.

With kind regards,

P.S. Click on the following link if you would like to recommend Nature Microbiology to your librarian <http://www.nature.com/subscriptions/recommend.html#forms>

** Visit the Springer Nature Editorial and Publishing website at http://editorial-jobs.springernature.com?utm_source=ejP_NMicro_email&utm_medium=ejP_NMicro_email&utm_campaign=ejP_NMicro for more information about our career opportunities. If you have any questions please click [here](mailto:editorial.publishing.jobs@springernature.com). **

We are grateful to the reviewers for their positive comments and feedback and have addressed individual points below.

Referee #1 (Remarks to the Author):

Stabell et al investigate pathways to neutralization resistance to two clinically used broadly neutralizing antibodies (bnAbs), 3BNC117 and 10174, in a large-scale study. They performed more than 7,000 parallel selection experiments for 15 primary isolates, spanning global diversity. Their conclusions are certainly of interest in understanding the range of mechanisms by which neutralization resistance is achieved to these two prototype bnAbs. Many of the mechanisms have been described earlier, although in less detail than here. The translational implications of the study are alluded to but not clearly explained.

The study demonstrates that the Envelope mutations providing resistance (defined as an $IC_{80} > 10$ $\mu\text{g/ml}$ or a 10-fold increase in IC_{80} over wild type) to the two bnAbs are variable and isolate dependent. I understand the $IC_{80} > 10$ $\mu\text{g/ml}$ but the 10-fold increase appears less useful since, if an isolate had a very low IC_{80} , then the 10-fold increase would still render the isolate sensitive to bnAb. In other words, the criterion might be interesting but would not have translational impact.

Response: The criteria, IC_{80} increase of 10-fold or $> 10 \mu\text{g/ml}$, were selected so as to include the maximum number of substitutions in our analysis. The vast majority of the substitutions selected resulted in acquisition of significant resistance with $IC_{80} \geq 25$ (Figures 3a-b). In the few remaining cases we would respectfully argue that an increase in IC_{80} from, for example, 0.22 to 4 would be considered significant as the concentration of antibody required to suppress viremia would be significantly higher than that necessary to suppress the parental virus.

Many of the mutations providing 3BNC117 resistance are known and are within the CD4 binding site. Others are more distant and are likely associated with conformational rearrangements although these are not structurally explored.

Response: Structural studies of individual amino acid substitutions are beyond the scope of this study. We further note that crystal structures of the envelope trimers of the strains used in our study are not available. Nevertheless, we have mapped substitutions onto previously characterized crystal structures in figures 6d and extended data 3e.

Some mutations were specific to 3BNC117 resistance, others provide resistance to multiple CD4 binding site bnAbs. As the authors point out in the Discussion, some of the mutations may lead to a more "open" state of Envelope and are unlikely to survive in vivo as they will be sensitive to weakly neutralizing/non-neutralizing Abs. They conclude that a combination of CD4 binding site Abs may be more effective than a single bnAb in vivo and I think they would find general agreement on this point and it has been advanced.

For bnAb 10-1074, resistance is much more closely focused on mutations that lead to the loss of the glycans at N332 or N334. Mutation at position 325 is also noted. Again, this is in line with much published literature.

An interesting observation of the paper, as above, is the variability of escape pathways open to HIV, dependent upon the bnAb and the isolate considered. A downside of this observation is the likelihood that one might need to carry out the type of study described for each bnAb and range

of isolates of interest. However, the authors did nicely carry out a bootstrap analysis to suggest that their suggest that their experiments had captured the majority of resistance sites in the panel of viruses tested.

Overall, this is an interesting study of bnAb resistance to HIV but probably best placed in say Nature Medicine than Nature.

Referee #2 (Remarks to the Author):

The manuscript by Stabell et al aims to define pathways to viral escape from two broadly neutralizing antibodies, 10-1074 and 3BNC117. Through an impressive number of parallel selection experiments, the authors show that there is substantial variability in mutations that confer resistance among isolates, and consistent with previous studies that single mutations within the epitope region can confer resistance. A particularly intriguing finding is that some mutations alter cell-cell transmission, and under explored aspect of bnAb resistance. Although this expands our understanding of the variability and bnAb escape and further highlights the risk that resistance poses to bnAb based interventions, it does not represent a fundamental shift in our understanding of bnAb escape.

Figure 1 – can the authors clarify the basis of strain selection?

Response: The goal was to select a minimum of two viral strains from each clade A, B, C, or D. The strain selection from within each of these clades was based on availability of replication competent infectious molecular clones and sensitivity to the bnAbs used in the study. This is now clarified in the Methods section, lines 506-508.

Line 81 – was there a formal analysis to show increased diversity for the antibody selection experiments, and where that was loc. Was that increased diversity consistent across strains?

Response: The reviewer raises an excellent point that was not made clear in the original manuscript. We sequenced all viral pools prior to selection. The vast majority of mutations were low-frequency and randomly distributed across the envelope coding sequence. We note that the limit of detection of variants in standard illumina amplicon sequencing is around 1 in 1,000 to 10,000 (i.e. $\geq 0.1-0.01\%$ of the reads). We have included a graph showing this data in extended figure 2a. A comparison with putative resistant mutations following selection shown in the graphs in figure 2b shows a specific enrichment across all strains. The majority of bnAb resistance mutations were present below the level of detection prior to selection, see also figure 1e.

Page 561 – how did the authors approach covariation of putative escape mutations? And what was the basis of the choice of double mutations?

Response: We are unclear about the place in the manuscript the reviewer is referring to. For all viruses, each putative resistance mutation was introduced individually into the wildtype envelope. In the majority of cases, a single amino-acid substitution was sufficient to explain the resistance observed (figure 3 c/d) and therefore, combinations with other changes were not tested. Two strains were the exception: TRO.11 and CH607. For TRO.11, all possible combinations of mutations identified in any isolate were combined. For CH607, only mutations identified in the same isolates, ie linked, were combined. Since the mechanism of escape of CH607 was via increased cell-to-cell spread further combinations of mutations were not deemed necessary.

Line 135 – did the authors consider mutations outside of the conventional epitope regions. This would be especially interesting for understudied subtypes.

Response: All putative mutations were identified computationally with no apriori knowledge of the known epitopes. This has been clarified in the revised manuscript, Results, line 133-134. We note that many putative mutations in TRO.11 and CH607 (for 3BNC117) and SC191727 (for 10-1074) were located outside of the known epitopes.

Line 181 – do the authors have any insights in why the genetic barrier appears to be higher for some strains – was it related eg to starting sensitivity or to viral “fitness”, here viral entry

Response: The starting IC80 value minimally correlated with starting susceptibility (as measured by IC80) for 10-1074. 3BNC117 showed essentially no correlation. See figures below. For unique situations like TRO.11, where multiple mutations are required for resistance, we believe there is no way to predict whether an envelope will have a high or low genetic barrier to resistance, and suggest that, at this point in time, only experimentation can reveal this.

Line 175 – was low-titer / dead virus particularly observed for low-frequency mutations and was it context dependent?

Response: For the strain in question, 9004SDM, our data shows that the V5 deletion was clearly responsible for escape. The primary focus of the study was to identify mechanisms of escape

and therefore we did not further explore individual amino acid substitutions. Additional adaptive changes during passage, ie prior to selection could potentially affect the impact of these individual changes in the context of the full-length virus but this was beyond the main goals of the study.

Line 195 – not clear whether changing criteria such as location of mutations would be valuable here?

Response: It is unclear to us what the reviewer means. This analysis was done by site, i.e. location of the mutation.

Line 231 – interesting that sensitivity increases – can the authors investigate this in more detail

Response: The sensitization to other bnAbs was in general minimal (<10-fold increase). We speculate these amino acid changes might slightly strengthen binding of other bnAbs, however the existing crystal structures for the envelope-bnAb pairs used in this study are not available and would be beyond the scope of this study.

Line 261 – there are quite a few more studies, including from natural history cohorts, AMP, etc

Response: We agree with the reviewer, however, there was a limit on the number of references allowed. Therefore, we only included studies that directly confirmed the impact of individual changes on bnAb sensitivity. Nevertheless, we have included additional references that we hope would be acceptable, discussion line 267 refs 28, 47-50.

Line 312 - unclear to me that is meant here

Response: We have clarified this point, Discussion, lines 319-320.

We appreciate the reviewer's comments and have addressed them in the manuscript. A detailed response is provided in red below.

Reviewers Comments:

Reviewer #1 (Remarks to the Author):

I was Referee #1 on the original submission of this manuscript to Nature so I will only make a few summary remarks and comments on the authors' response.

The original paper described a well conducted study that I considered fell a little below the level of impact to make it into Nature. However, it is an important study and well worthy of publication.

Two comments on the authors' response:

1. The authors did not respond to my comment about the translational studies being alluded to but not clearly explained. Here, I understood that the authors envisaged translation: "These data provide a rationale for selecting bnAb combinations that are most likely to achieve treatment success" and "Nevertheless, approaches such as those documented here in can identify combinations of bnAbs with a higher barriers to escape that could guide strategies to minimize the development of resistance". I was looking for the authors, given their experiences in this study, to provide a roadmap for the field about how this might be achieved. Which viruses to use, how many, what conditions etc etc.

We have included a more detailed outline, based on our data (and experience), that could facilitate identification of 'resistance proof' bnAb combinations in the discussion, **page 9, lines 292-305**.

2. For the structural comment, I was interested to see mapped on a (gp120?) structure where the significant mutations distant from the CD4 binding site were. I did not really get that from the two figures described.

We have included a new supplementary figure, **page 33, extended data figure 4**, to highlight the amino acid substitutions that confer resistance to 3BNC117 but lie outside the CD4 binding site.

Again, a fine manuscript-the above are suggestions for improvement.

Reviewer #2 (Remarks to the Author):

This is a well-done and timely manuscript delving into escape pathways from bNabs 3BNC117 and 10-1074. These data are highly clinically relevant. The authors addressed all the reviewers comments.

I have minor comments.

In response to Reviewer 2's point about Line 195, I wonder if the reviewer meant regions of the

mutations (ie, did almost all resistance mutations arise in Loop D or V5 for 3BNC117 and glycan 332 for 10-1074 as it looks like they might from Figure 2a and b?)

The reviewer is correct, the majority of confirmed resistance sites arose at loop D (275-282) or V5 loop (457-470) for 3BNC117 and V3 glycan epitope (332-334) for 10-1074 (see Figure 3a,b). This is stated in the results, **page 4, lines 117-124**.

On Line 240, 10-1074 and PGT121 not only target the same glycan epitope, they are clonal relatives, inferred to have derived from the same ancestral naive B cell (I think?).

The reviewer is correct, PGT121 and 10-1074 are clonal relatives isolated from the same individual. They differ only in their pattern of somatic hypermutation. We clarified this point and added the original reference to the results section, **page 7, line 214**.

Reviewer #2 (Remarks on code availability):

I have no expertise to assess code.

Reviewer #3 (Remarks to the Author):

Summary

This manuscript by Stabell et al. investigates HIV-1 escape mutations from the broadly neutralizing antibodies 10-1074 and 3BNC117 using a replication-competent viral system. While escape mutations for these bnAbs were recently mapped by Radford et al (JV, 2025). using two divergent Env strains, the present study extends this work by analyzing resistance across 15 primary HIV-1 isolates and assessing cross-resistance against an additional panel of nine bnAbs. This expanded scope provides a broader experimental basis for understanding resistance mechanisms and for informing bnAb combination strategies.

The experimental and analytic approaches are well thought out. In particular, resistance mutations arise through natural Env diversification during multi-cycle replication and are selected in a 96-well bottlenecked format, where resistant viruses typically expand from a single dominant lineage. This strategy can reduce several technical confounders associated with bulk selection or single-round-infection of mutant libraries. The resistance scoring framework, which integrates unselected mutation frequency with recurrence across independent resistant isolates, is reasonable and conceptually sound.

Overall, this is a strong and complementary study with clear relevance to bnAb therapeutic design, although several aspects of contextualization and data presentation could be improved.

Major Concerns

1. The manuscript does not adequately discuss recent work by Radford et al. demonstrating strain-dependent escape pathways for 10-1074 and 3BNC117.
 - The statement (line 55) that it is “unknown whether HIV-1 isolates differ in their ability to escape bnAb neutralization” is inaccurate and should be revised.
 - A clearer comparison explaining how this study extends or complements prior escape-mapping work would strengthen the framing and help avoid overstatement of novelty.

We respectfully disagree with the reviewer. The paper by Radford et al, cited in the introduction of the originally submitted manuscript, investigated escape pathways using deep mutational scanning (DMS) with only 2 strains. In our view this does not constitute a significant expansion over previous studies. The inclusion of 9-15 strains spanning the global HIV-1 diversity used in our experiments is novel and so is our approach that allows for the identification of resistant escape pathways that would be missed by the DMS approach. This is illustrated by a direct comparison between the two approaches; both studies have examined 3BNC117 resistance in the context of strain TRO.11. The DMS approach identified individual amino acid changes that increased resistance to 3BNC117 by less than 8-fold. Our study revealed that the escape mechanism of TRO.11 is unusual and requires a combination of amino acid changes to achieve near-complete resistance. Moreover, we identified multiple individual amino acid changes in the context of other strains that increase resistance to 3BNC117 by over 2 logs. Similarly, neither mutagenesis of strain BG505 nor DMS of strain BF520 succeeded in identifying individual amino acid substitutions that confer resistance to another CD4 binding site targeting antibody 04-A06 (Geiselman et.al., 2025). Although our study did not use the same strains, we have identified individual amino acid substitutions that confer resistance to 04-A06, despite not explicitly selecting for resistance to this bnAb (Fig. 6). We did not consider it was necessary to discuss the limitations of the DMS approach at length in our manuscript and regarded the difference in scope between the two studies evident. Nevertheless, since the reviewer has requested this comparison, we have included a paragraph on the advantages and novelty that our approach offers over prior attempts at mapping resistance in the discussion, **pages 9, lines 269-285** and have modified the statement in **page 2, lines 47-49** for accuracy. The comparison shows that certain questions cannot be adequately addressed by site-directed mutagenesis-based approaches and that experimental virus evolution provides a more comprehensive answer.

2. Resistance mutation mapping and sequencing quality remain unclear.

- The authors used Nextera XT–based Illumina sequencing to profile mutations in ~2.5 kb env amplicons, but sequencing quality metrics (e.g., read depth and coverage) were not provided. Given that this approach can produce uneven coverage, conclusions regarding single- versus multi-site resistance mutations are difficult to evaluate technically without additional sequencing QC data; “not observed” may reflect true biological absence of mutations but may also result from lack of detection.

We have provided metrics and a more detailed explanation of the mapping approach in the methods section, **pages 17-18, lines 606-615**. As noted in this section, we did not analyze any variants that accounted for less than 25% read frequency, mitigating the requirement for deep sequencing.

- The resistance scoring framework incorporates mutation frequencies from the “pre-selection population,” but this term is ambiguous. It is unclear whether it refers to virus populations at an early stage of the experiment (e.g., virus produced from transfected cells or after the first passage) or to viruses from the control plates that underwent several passages without selection. If the latter (which likely the case), a clearer term (e.g., “no-selection population”) would improve clarity.

We have clarified these questions in the methods section, **page 18, lines 610-615**. Two control populations were sequenced: the pre-selection population and the “no-selection population” (i.e.

virus that underwent an identical experimental procedure as the selected viruses but without bnAbs presence). The mutational patterns between these populations were very similar and were used in combination or interchangeably.

- In addition, it is unclear how the control plates were handled prior to sequencing (e.g., number of passages, duration of culture).

The control plates were handled identically to the experimental plates, i.e. virus was grown until $\geq 40\%$ of the cells were infected but in the absence of bnAbs. We have included this clarification in the methods section, **page 16, lines 560 and 571-573**.

- While the scoring framework is reasonable, the manuscript would benefit from a clearer discussion of its assumptions, limitations and significance, particularly in cases where multiple mutations are present; and in comparison to other approaches using bulk /single-infection selection of mutant libraries.

As mentioned above we have included a new paragraph in the discussion, **page 8-9 lines 269-285**, that discusses more extensively the significance and advantages of our approach over prior approaches.

Minor Concerns

- Clarify the frequency thresholds used to define dominant resistance mutations within individual wells.

We have clarified this in the methods, **page 17-18, lines 609-619**.

- Indicate how often multiple resistance mutations were observed relative to single-mutation outcomes.

Less than 5%, 20/406, of 3BNC117 resistant isolates had more than one confirmed resistance mutation (i.e. the majority of selected wells had only a single resistance mutation identified). This is now stated in **page 5, lines 146-147**.

- Specify whether any regions of env were excluded from analysis due to poor coverage or alignment ambiguity.

All extracellular envelope regions were included in the analysis, this is clarified in the methods, **page 17, lines 606-607**.

- Consider adding a schematic summarizing the experimental workflow of the resistance scoring system (including the role of the control plates) to improve clarity.

We have included a schematic of our scoring method and validation of the algorithm-selected mutations via pseudotyped neutralization assays in **page 32, extended data figure 3**.

We are grateful to the reviewer for his comments and pleased that major and minor issues have been addressed satisfactorily. We have deposited our datasets on publicly accessible websites as requested and noted in the author checklist.

Reviewer #3:

Remarks to the Author:

Overall, the revised version addresses the main technical and contextual concerns raised in the previous review.

The authors have revised the previous statement “It is unknown whether HIV-1 isolates differ in their ability to escape bnAb neutralization” to “Evaluation of bnAb escape across a panel of HIV-1 primary isolates that span global viral diversity has not been previously attempted, and it is unknown whether there is a subtype/geographic variation in the propensity of HIV-1 to acquire bnAb resistance,” thereby emphasizing global testing across a panel of HIV-1 primary isolates in this study, and have added a clearer discussion comparing their approach with prior deep mutational scanning studies. The revised manuscript text more appropriately contextualizes novelty and scope, and I consider this issue adequately addressed.

Concerns regarding sequencing quality and resistance mutation mapping have been clarified through added methodological detail, including explicit variant frequency thresholds, reporting of average read depth, handling of control populations, and justification for focusing on dominant variants arising from bottlenecked selections. Given the experimental design, these clarifications render the conclusions technically reasonable.

Data transparency: The revised manuscript reports only the average read depth of 930 reads/nucleotide for the 2.5-kb env amplicon. Public deposition of the raw (or minimally processed) sequencing data in a standard repository—as a reference dataset for future studies—would allow interested readers to reproduce the analysis and independently evaluate read-depth distribution and coverage uniformity. I encourage the authors to make these data publicly accessible to strengthen transparency and reproducibility.

Ambiguities surrounding the definition and handling of control populations are now resolved, and the added discussion of the resistance-scoring framework and its assumptions strengthens the manuscript.

All minor points have been satisfactorily addressed.